# Pre-training Auto-regressive Robotic Models with 4D Representations

**Dantong Niu** [* 1]  **Yuvan Sharma** [* 1]  **Haoru Xue** [1]  **Giscard Biamby** [1]  **Junyi Zhang** [1]  **Ziteng Ji** [1]
**Trevor Darrell** [† 1]  **Roei Herzig** [† 1]

https://arm4r.github.io/

## Abstract

Foundation models pre-trained on massive un-labeled datasets have revolutionized natural language and computer vision, exhibiting remarkable generalization capabilities, thus highlighting the importance of pre-training. Yet, efforts in robotics have struggled to achieve similar success, limited by either the need for costly robotic annotations or the lack of representations that effectively model the physical world. In this paper, we introduce ARM4R, an **A**uto-regressive **R**obotic **M**odel that leverages low-level **4D R**epresentations learned from human video data to yield a better pre-trained robotic model. Specifically, we focus on utilizing 3D point tracking representations from videos derived by lifting 2D representations into 3D space via monocular depth estimation across time. These 4D representations maintain a shared geometric structure between the points and robot state representations up to a linear transformation, enabling efficient transfer learning from human video data to low-level robotic control. Our experiments show that ARM4R can transfer efficiently from human video data to robotics and consistently improves performance on tasks across various robot environments and configurations.

## 1. Introduction

Recently, foundation models (FMs) have shown remarkable success, particularly in the domains of language (Brown et al., 2020; Touvron et al., 2023), vision (Kirillov et al., 2023), and multi-modal models (Chen et al., 2022; Alayrac et al., 2022; Liu et al., 2023; Li et al., 2023; OpenAI, 2023) pre-trained on vast amounts of vision and text data. These

---
[*]Equal contribution. [†]Equal advising. [1]Berkeley AI Research, UC Berkeley. Correspondence to: Dantong Niu <bias_88@berkeley.edu>.

*Proceedings of the 42nd International Conference on Machine Learning*, Vancouver, Canada. PMLR 267, 2025. Copyright 2025 by the author(s).

models exhibit impressive zero-shot and few-shot learning capabilities (Radford et al., 2021; Ouyang et al., 2022; Wei et al., 2021; Chung et al., 2024), highlighting the power of pre-training on generic data. However, numerous attempts in robotics (Xiao et al., 2022; Kim et al., 2024; Zhen et al., 2024b; Niu et al., 2024; Ye et al., 2024) have yet to achieve the same pre-training success seen in other domains. This could potentially be attributed to the scarcity of large-scale, diverse robotic data, unlike the abundance of text and image data available for vision and language FMs.

The lack of robotic data poses a significant bottleneck in training foundation models that can effectively generalize across diverse robotic platforms and tasks. To overcome this limitation, several recent approaches (Xiao et al., 2022; Ye et al., 2024) employ representation learning by pre-training on an abundance of human data, enabling transfer to robotic systems. These approaches aim to recognize the inherent similarities between human and robot manipulation tasks and exploit the vast repositories of human video data available on the internet. Yet, these approaches have not been able to demonstrate effective generalization to downstream tasks. In part, this is due to their representations lacking an understanding of the physical world (Zhen et al., 2024a), and therefore being less effective for robotics.

In contrast with these methods, Vision-Language-Action (VLAs) models take a slightly different approach, implicitly leveraging human data in robotics by incorporating pre-trained components from Vision-and-Language Models (VLMs). In particular, they use language decoders pre-trained on tasks like visual question answering (e.g., RT-2 (Brohan et al., 2023a)) and image captioning (e.g., Open-VLA (Kim et al., 2024)). Despite such efforts, there is a discrepancy between these models' high-level pre-training objective and the goal of enabling robotic models to handle low-level action prediction. While these initial objectives are valuable for comprehending visual and linguistic content, they don't directly address the nuances of low-level robot control, which involves aspects like precise manipulation and spatial reasoning. To address this, this paper's method employs a lower-level pre-training objective by starting with a model that utilizes next-token prediction to learn 4D representations from human video data. These representations

## Pre-Train on 4D Representations from Human Video Data

Figure 1: **Overview of ARM4R.** We introduce an **A**uto-regressive **R**obotic **M**odel that leverages low-level **4D R**epresentations (3D point tracks across time) learned from human videos to yield a better pre-trained robotic model.

can then be transferred to more specialized scenarios by fine-tuning on robotic scenes and subsequently on proprioceptive data, while maintaining the same training objective.

In this paper, we introduce ARM4R (**A**uto-regressive **R**obotic **M**odel with **4D R**epresentations).[1] The key insight behind ARM4R is to learn a low-level representation from the abundance of human video data that can capture properties of the physical world. This involves lifting 2D representations to 3D using monocular depth estimation and subsequently tracking the 3D points. The resulting 4D representations maintain a shared geometric structure — up to a linear transformation — between the 3D points and robot state representations used downstream, enabling efficient transfer learning from human video data to robotic manipulation tasks. Surprisingly, pre-training our method solely on human data yields superior results compared to other models like VLAs (Kim et al., 2024) that are pre-trained on robotic data such as OpenX (Collaboration et al., 2023).

We summarize our main contributions as follows: (i) We introduce a novel robotics pre-training approach that incorporates low-level 4D representations that enhance understanding of the physical world while also learning from unlabeled videos. (ii) Our approach shows that pre-training solely on human video data can lead to better performance than other methods that are pre-trained only on robotic data; (iii) Our

method on average surpasses baselines like PerAct (Shridhar et al., 2023) on RLBench, and LLARVA (Niu et al., 2024), OpenVLA (Kim et al., 2024), and $\pi_0$-FAST (Pertsch et al., 2025) on real tasks with a 7-DoF Kinova Gen3 robot; (iv) Our model also exhibits several advantageous properties, including cross-robot generalization and 3D point track prediction for out-of-domain human and robotic videos.

## 2. Related Work

**Vision-Language-Action Models**. VLAs are a type of robotic model that combines visual perception, language understanding, and action generation capabilities. VLAs take as input visual observations along with a language instruction, and output a sequence of robot control actions. Several VLAs, such as LLARVA (Niu et al., 2024), OpenVLA (Kim et al., 2024), LLaRA (Li et al., 2024a), and RoboPoint (Yuan et al., 2024) directly fine-tune a VLM to predict robot actions, often using special tokens to represent the action space. These models differ in the choice of VLM and the specific method used to encode robot actions, but they share the underlying principle of adapting a pretrained VLM for robotic control. A similar model is 3D-VLA (Zhen et al., 2024a), which consists of components for generating future states of an environment based on data that includes 3D information, such as point clouds. These existing VLAs utilize language decoders that have been pre-trained for high-level tasks like image captioning (Kim et al., 2024)

---

[1] ARM4R is pronounced "armor".

and VQA (Brohan et al., 2023b), which may be inadequate for low-level robotic environments. In contrast, we show that leveraging low-level vision representations from human video data can result in a better pre-trained robotic model.

**3D Motion Fields**. Motion estimation spans from 2D optical flow (Horn & Schunck, 1981) and object tracking (Wu et al., 2013) to recent dense point tracking (Harley et al., 2022). Moving from 2D to 3D further enriches the geometric understanding. Early work on scene flow (Vedula et al., 1999) estimates short-term 3D motion based on explicit 3D structure (Menze & Geiger, 2015) or depth images (Teed & Deng, 2021). More recently, SpatialTracker (Xiao et al., 2024) tackles long-range 3D point tracking by lifting 2D pixels into 3D with monocular depth estimates and iteratively refining 3D trajectories with as-rigid-as-possible motion priors. This 3D-driven strategy greatly improves occlusion robustness and yields impressive 3D point tracking results.

In robot learning, 2D motion fields have been used to enable fine-grained control, guiding manipulation and imitation learning (Goyal et al., 2022; Vecerik et al., 2023; Gu et al., 2023; Yuan et al., 2024; Zheng et al., 2024; Bharadhwaj et al., 2024; Xu et al., 2024). Despite their success, these approaches remain limited by the lack of geometric cues and less spatial awareness. In contrast, 3D motion fields offer more spatially grounded representations, enabling more efficient policy learning. ToolFlowNet (Seita et al., 2022) leverages scene flow to estimate tool trajectories in behavior cloning, though it uses only a relatively coarse 3D signal. We instead adopt dense 3D point tracking on diverse human videos, and use these rich 4D representations (3D points tracked across time) to pre-train a general auto-regressive robotic model with robust and versatile action generation.

RVT (Goyal et al., 2023) and RVT-2 (Goyal et al., 2024) are recent transformer-based methods that use a more direct approach for learning 3D information, leveraging ground-truth RGB-D images to reconstruct a scene's point cloud and then predict keyframes. This point cloud reconstruction relies on a similar principle to our method, with ARM4R instead using pre-training on 3D point tracking to learn scene structure.

**Pre-training for Robotic Models**. Pre-training has emerged as a crucial technique for improving the performance and generalization capabilities in robotics. Large-scale datasets such as OpenX (Collaboration et al., 2023) contain diverse sensor modalities, tasks and action spaces across various robots. Models trained with these datasets, such as RT-1-X (Brohan et al., 2023c), RT-2-X (Brohan et al., 2023a), Octo (Team et al., 2024), OpenVLA (Kim et al., 2024) and LLARVA (Niu et al., 2024), can be applied in various robot embodiments and tasks. Yet, these robot pre-training datasets are still orders of magnitude smaller than the data that current LLMs and VLMs are trained on.

To address the data issue, another prominent pre-training approach is to leverage large-scale datasets of human videos. This harnesses the abundance of freely available human activity data on the internet, offering a scalable alternative to collecting expensive robot demonstrations. For example, Track2Act (Bharadhwaj et al., 2024) trains a 2D point-tracking model on human videos from Epic-Kitchens100 (Damen et al., 2018) and Something-Something-v2 (Goyal et al., 2017), then re-purposes it to guide robotic manipulation. Any-Point Trajectory Modeling (ATM) (Wen et al., 2024) similarly utilizes a small set of human demonstrations to aid cross-embodiment transfer, though in a more task-specific setting and still relying on 2D motion. By contrast, our approach lifts 2D observations into 4D representations (3D plus time), which not only enhances spatial awareness and occlusion handling, but also allows pre-training on human videos at scale, providing broader applicability and more robust policy learning in robotics.

## 3. Auto-regressive Robotic Models

To address the challenge of leveraging pre-trained vision representations from human video in robotic models, we present an auto-regressive model that relies on low-level 4D representations. The model is trained in three stages. The first stage—the pre-training stage—focuses on learning generalized low-level representations through 3D point tracking from human videos. In the second stage, the model is fine-tuned for the same task, but using a small amount of data for the robot that we intend to use in downstream tasks. Finally, the third stage fine-tunes the model for robotic control. We begin by discussing the preliminaries (Section 3.1), then introduce the architecture (Section 3.2), and the training procedures (Section 3.3). Our method is shown in Figure 2.

### 3.1. Preliminaries

**4D Scene Representations**. Our 4D representations result from solving the 3D point tracking problem, which involves finding the 3D coordinates of discrete points across time, given a monocular video consisting of $T$ discrete frames. Formally, the objective is to find $p_t$ as defined below:

$$p_t = \{(x_{jt}, y_{jt}, z_{jt}) \mid 0 \leq j < n\} \qquad (1)$$

where $n$ is the total number of points being tracked, and $0 \leq t < T$. In solving this tracking problem, the identities of the points are fixed and consistent across all frames: the $j$-th point in $p_t$ refers to the same physical point in 3D space across all time steps $t \in [0, T)$. To initialize these points, we define a square grid of size $g \times g$ on the first frame (frame $t = 0$), resulting in $n = g^2$ points. The task is to track the 3D coordinates of these initial queried $n$ points throughout the video while maintaining their unique identities.

**4D Representations for Robotic Manipulators**. One of

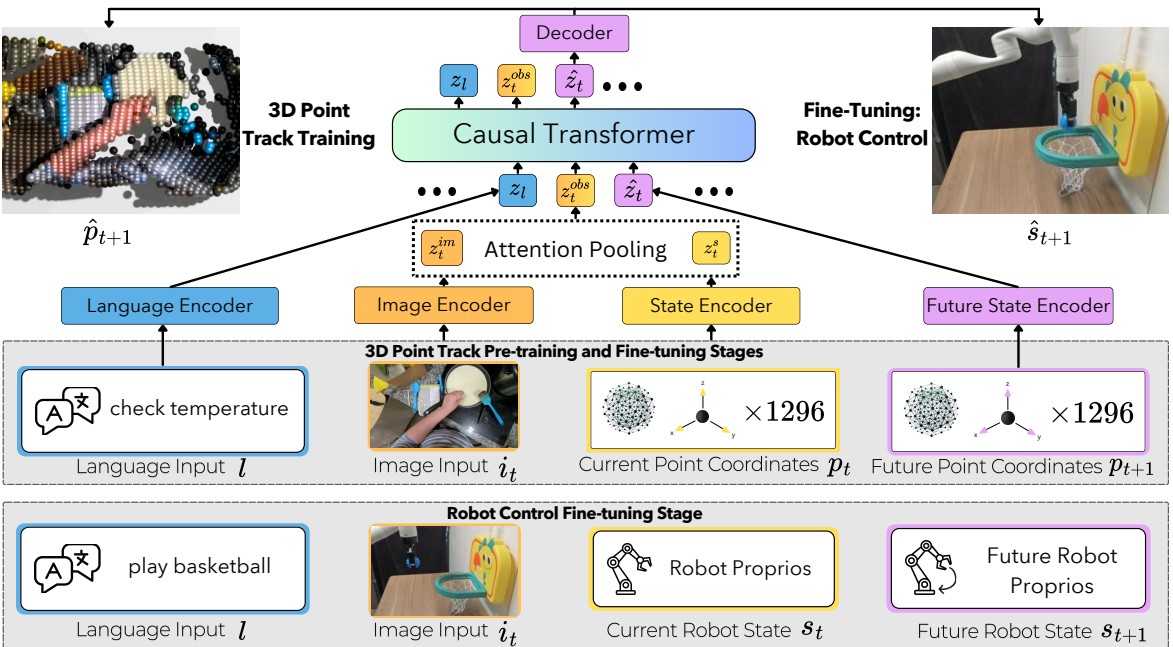

Figure 2: ARM4R is trained in three stages. **Top Grey Box**: The first two stages focus on learning a scene-wide 4D representation by predicting 3D points across time, where Stage 1 pre-trains on a large egocentric human dataset (Epic-Kitchens100), and Stage 2 fine-tunes on a smaller dataset (1-2K demonstrations) of robotic scenes, adapting the point tracking to robotic scene and camera. **Bottom Grey Box**: Finally, the model is fine-tuned to predict robot proprioceptive states rather than 3D points to enable robotic control.

the main benefits of using 4D representations is that the 3D point tracks in a robotic setting are described by linear transformations constructed using the robot states. We now present a simple proof of this claim.

Consider a $n$-DoF open-chain manipulator, initially in its reference configuration $\theta_1 = 0, \theta_2 = 0, \cdots, \theta_n = 0$. Suppose the manipulator is commanded to a new position $\theta_1 = \theta_1^d, \theta_2 = \theta_2^d, \cdots, \theta_n = \theta_n^d$. Next, we take any arbitrary point $p$ on the body of the manipulator lying between joints $i, i+1$. Let $p$ be described by the transformation $g_{i,p}(0)$ in joint $i$'s frame. Then, $p$'s new position in the base frame can be described by:

$$g_{1,p}(\theta) = e^{\hat{\xi}_1 \theta_1^d} \cdots e^{\hat{\xi}_i \theta_i^d} g_{i,p}(0)$$

where $\hat{\xi}_j$ are the twists associated with each joint. Since this product of 4x4 SE(3) matrices represents a linear transformation, points on the robot body as well as attached rigid bodies evolve through linear transformations in terms of the robot states. Additionally, for overactuated manipulators (7-DoF), the robot learning problem simplifies to tracking arbitrary trajectories, as configuration space limitations are less restrictive. This allows the model to benefit more effectively from the 3D point tracking pre-training task.

**Robotic Episodes**. Robotic control can be formulated as a finite-horizon Markov Decision Process (MDP), character-

ized by temporal sequences that capture the robot completing a particular task. The task is described by the language instruction $l$. The temporal sequences typically consist of visual observations $i_{0:T-1}$ and proprioceptive states $s_{0:T-1}$, which can lie in Cartesian space or joint position space. Then, the objective is to learn a policy that predicts one or more future actions, conditioned on a finite number of previous timesteps, to successfully complete a given task.

**Inputs**. Given any video, we structure our models' input at timestep $t$, into three parts: the language instruction $l$, the image input $i_t$, and the current 3D coordinates of the tracked points, $p_t$. These elements together provide the contextual, visual, and spatial information necessary for 3D point tracking. The output is the future 3D coordinates of the tracked points, $p_{t+1}$. When fine-tuning the model for robotic control (see Section 3.3), we replace the tracked input points with the robot's current state $s_t$, and the output points with the next state, $s_{t+1}$. We hypothesize that the shared geometric structure — up to a linear transformation — between the points and robot state representations enables efficient transfer learning between the second and third stages.

### 3.2. Architecture

In the first and second training stages, our objective is to develop an auto-regressive model $\pi$ capable of predicting

3D point tracks. The predictions are conditioned on the input $(l, i_t, p_t)$ from a context window of $C$ timesteps:

$$\pi(l, i_{t-C+1:t}, p_{t-C+1:t}) \rightarrow p_{t+1} \qquad (2)$$

During control fine-tuning, the objective changes slightly, as the model conditions on, and predicts proprioceptive states:

$$\pi(l, i_{t-C+1:t}, s_{t-C+1:t}) \rightarrow s_{t+1} \qquad (3)$$

Before being fed into the causal transformer for next-token prediction, each part of the input and output must be processed and projected into the same latent space. To achieve this, separate encoders are used for the language, image, points, and robot states, as we discuss below.

**Language Encoder**. We use a frozen CLIP (Radford et al., 2021) text encoder trained on LAION-2B (Schuhmann et al., 2021) to process text, with a learnable linear projection layer added at the end to get the language token $z_l$.

**Image Encoder**. To process the image, we use a standard Vision Transformer (Dosovitskiy et al., 2021) to get the image token $z_t^{im}$. This ViT is frozen while training our model, and is pre-trained using CrossMAE on a combination of ImageNet (Deng et al., 2009) and the OpenX dataset (Collaboration et al., 2023). This enables the vision transformer to learn to encode both non-robotic and robotic data, which is important since our pre-training stage emphasizes the former, while fine-tuning targets the latter.

**4D Representations**. To encode the point coordinates, we use a standard 2-layer MLP. The resulting feature $z_t^{pts}$ is combined with $z_t^{im}$ via an attention pooling layer to get the current observation token, $z_t^{obs}$. A separate MLP is used to encode the next timestep's point coordinates and get $\hat{z}_t$.

**Causal Transformer**. For each timestep, we get three tokens, one each for language, current visual observation and prediction, which are fed into the transformer. In our implementation, we use a randomly initialized causal Transformer (ViT-Base). The Transformer is trained for standard next-token prediction on the sequence $(z_l, z_t^{obs}, \hat{z}_t, z_l, z_{t+1}^{obs}, \hat{z}_{t+1}, \cdots)$, with loss only being calculated for $\hat{z}_t$. During inference, we input $(z_l, z_0^{obs})$ at timestep 0, and the model predicts $\hat{z}_t$ for every timestep.

**Decoder and Loss Function**. We calculate the loss using only $\hat{z}_t$, as predicting $z_l$ and $z_t^{obs}$ is not the objective in either 3D point tracking or robotic control tasks. The predicted token is decoded using a two-layer MLP into the predicted point tracks $\hat{p}_{t+1}$. The L1 distance between $\hat{p}_{t+1}$ and the ground truth point tracks $p_{t+1}^*$ is used as the final loss:

$$\mathcal{L}(\hat{p}_{t+1}, p_{t+1}) = \frac{1}{n}\|\hat{p}_{t+1} - p_{t+1}^*\|_1 \qquad (4)$$

**Adaptation for Robotic Control**. We note that when fine-tuning for robotic control, we replace the MLPs for processing points $p_t$ with similar MLPs for processing robot states $s_t$. Additionally, when processing multiple images in the fine-tuning stage, we combine the observation tokens by concatenating linear transforms of the different views to get a single $z_{obs}^t$ token. The model is also trained to predict multiple future proprioceptive states. The rest of the architecture is kept the same (e.g. loss function). For more details, please see Appendix C.1.

### 3.3. Training

As previously mentioned, ARM4R is trained in three stages: the first two stages focus on the 3D point tracking task for human and robot videos respectively, and the last stage focuses on robotic control. Next, we describe these stages.

**Stage 1: Human Videos Pre-training**. In the pre-training stage, we focus on learning 3D point tracking, since this task allows our model to leverage large-scale human video data with a representation that also transfers over to the robotic domain. Specifically, we train our model on 76K videos from the Epic-Kitchens100 dataset (Damen et al., 2018), which contains rich human-object interactions with 97 verbs and 300 noun classes. By training to predict 3D point tracks for such large-scale human data, ARM4R gains a deeper understanding of the spatial dynamics and physical interactions of different bodies and objects, knowledge that is critical for enhancing robotic models.

To extract pseudo-annotations for 3D point tracks, we use an off-the-shelf tracker that generates 3D tracks for points arranged on a $g \times g$ grid. Points on the grid are initialized in the first frame of the video and tracked throughout the sequence. We note that the pseudo-labeled tracks are generated in the camera coordinate frame, inherently capturing both object and camera motion due to the egocentric nature of the human videos. In contrast, our robotic applications typically involve stationary cameras and different object-hand interaction patterns, introducing discrepancies in both camera dynamics and embodiment. To reconcile these differences and ensure smooth transfer to robotic domains, we introduce a fine-tuning stage focusing on 3D point tracking in the downstream robotic setup.

**Stage 2: Fine-tuning on 3D Point Tracking for Robotic Settings**. After the pre-training stage on human video data, we fine-tune ARM4R for the same 3D point tracking task with videos from the robotic setup we use in the downstream application. We note that this fine-tuning only needs to be performed once for every robot setup for all tasks combined, with a modest amount of data ($\approx 5-10\%$ compared to Stage 1). This step helps transition from the camera dynamics and embodiment gaps between the human video pre-training and the control fine-tuning in the next stage.

**Stage 3: Fine-tuning for Robotic Control**. Having trained

Table 1: **Success rate (%) on RLBench Multi-Task setting.** We compare ARM4R's performance against several related baselines on 12 tasks from the RLBench benchmark. We use 25 episodes per task and 5 random seeds, averaging the results to get the success rate. ARM4R achieves the best performance on 4 of 12 tasks and the best average success rate.

| Method | Task | | | | | | | | | | | | Average |
| | open drawer | meat off grill | turn tap | put money | push buttons | sweep dustpan | slide block | close jar | screw bulb | place wine | reach and drag | stack blocks | Success Rate (%) |
|---|---|---|---|---|---|---|---|---|---|---|---|---|---|
| Image-BC (ViT) | 0 | 0 | 16 | 0 | 0 | 0 | 0 | 0 | 16 | 0 | 0 | 0 | 2.67 |
| C2FARM-BC | 20 | 20 | 68 | 12 | **72** | 0 | 16 | 24 | 8 | 18 | 24 | 4 | 23.83 |
| ManiGaussian | 76 | 60 | 56 | - | 20 | 64 | 24 | 28 | - | - | **92** | 12 | 48.00 |
| LLARVA | 60 | 80 | 56 | 44 | 56 | **84** | **100** | 28 | 8 | 12 | 52 | 0 | 48.33 |
| PerAct | 80 | 84 | **80** | 44 | 48 | 56 | 72 | **60** | **24** | 12 | 68 | **36** | 55.33 |
| ARM4R | **88.8** | **94.4** | 61.6 | **92.0** | 67.2 | 72.0 | 85.6 | 24.0 | 10.4 | **36.0** | 77.6 | 4.0 | **59.47** |

the model on 3D point tracking, we then fine-tune it for robotic control. In this stage, we collect a number of robotic demonstrations depending on the downstream tasks. We note that we use significantly fewer demonstrations for real robotic tasks than other baselines (See Section 4.3). After collecting successful data of the robot performing the target task, we replace the current and predicted point tracks in the training process with current and predicted robot states.

## 4. Experiments and Results

We evaluate ARM4R on 12 tasks in RLBench (James et al., 2020) and compare to relevant 2D and 3D baselines. We also test and ablate our model on two real robots: a 7-DoF Kinova Gen3 robot, and a 7-DoF Franka Emika Panda robot.

### 4.1. Implementation Details

ARM4R is implemented using PyTorch (Paszke et al., 2019). We use ViT-Base as our vision encoder, which is pretrained as described in Section 3.2. We use SpatialTracker (Xiao et al., 2024) as our off-the-shelf 3D point tracker. We note that the model uses a maximum context window $C$, which is the number of previous timesteps it considers when predicting the next action. In practice, we use $C = 16$ for most tasks, increasing it to $C = 32$ for some long-horizon tasks (details in Appendix D). The model is also trained to predict the next 16 actions, but we only execute the first prediction during evaluation. In both our simulation and real settings, we use end-effector control, with the model predicting the Cartesian position and rotation of the end-effector, and a binary value for the gripper. Finally, we use 4 NVIDIA A6000 GPUs for training and a single NVIDIA A6000 GPU for evaluation. More information, like training and fine-tuning recipes, is in Appendix C.2.

### 4.2. Simulation Evaluation

**Experimental Setup**. We evaluate ARM4R on 12 RLBench tasks, and follow the settings in PerAct (Shridhar et al.,

2023). A task is defined as a collection of demonstrations of the robot interacting in a given scene, with object variations (such as color or size). We train ARM4R for each task using 190 successful demos for every variation of the task (for more details, see Appendix D), and evaluate using 25 episodes per task in the validation set. Every episode is scored either 0 for failure or 100 for success. We use 5 seeds, which are averaged to get the final success rate.

**Baselines**. We compare to several baselines for our simulation evaluation. Image-BC (ViT) is a 2D language-conditioned baseline model that uses a ViT vision encoder, reported in PerAct (Shridhar et al., 2023). To compare against two different methods that use 3D representations, we select C2FARM-BC (James et al., 2022) and PerAct, which use voxels as 3D input to calculate robot actions. To compare to a method with 3D temporal tracking similar to ours, we evaluate against ManiGaussian (Lu et al., 2025), which uses a dynamic Gaussian splatting representation to predict robot actions. Lastly, LLARVA (Niu et al., 2024) is a recent state-of-the-art VLA that directly predicts low-level robot actions given an image and proprioceptive information as part of a language prompt.

**Results**. We report our simulation results in Table 1. ARM4R achieves the highest average success rate across all the tasks, and the best success rate for 4 out of 12 tasks. In particular, ARM4R surpasses PerAct, which directly uses voxel information from the simulation environment as input. This approach is not scalable since voxel data is expensive to collect in the real world. Instead, ARM4R learns to model the 3D world by pre-training on 3D point tracking, and the impressive performance highlights the model's strong grasp of physical understanding. We also note that ARM4R's superior performance compared to LLARVA — a VLA which uses a pre-trained language decoder — emphasizes the effectiveness of our representation and pre-training approach.

**Failure Cases Analysis**. We also found some abnormal behaviors when evaluating ARM4R on RLBench, which can be further summarized into two categories. (1) Un-

Table 2: **Success rate (%) on the real Kinova Multi-Task setting**. We compare ARM4R's performance to four related baselines on 13 real tasks grouped into five categories. We use 25 episodes per task for evaluation, averaging the results over 3 seeds to get the final success rate. ARM4R outperforms both baselines on all the tasks.

| Method | pick cube up | | | destack | | stack | |
| --- | --- | --- | --- | --- | --- | --- | --- |
| | yellow | cyan | green | yellow | cyan | yellow on cyan | cyan on yellow |
| ATM | $5.3 \pm 3.5$ | $6.7 \pm 2.7$ | $9.3 \pm 1.3$ | $4.0 \pm 2.3$ | $9.3 \pm 3.5$ | $1.3 \pm 1.3$ | $2.6 \pm 1.3$ |
| LLARVA | $44.4 \pm 6.4$ | $41.6 \pm 4.2$ | $54.2 \pm 11.0$ | $8.3 \pm 4.8$ | $10.3 \pm 2.6$ | $5.6 \pm 2.8$ | $12.8 \pm 2.6$ |
| $\pi_0$-FAST | $63.0 \pm 3.7$ | $33.3 \pm 8.3$ | $25.0 \pm 0.0$ | $5.5 \pm 2.8$ | $25.6 \pm 2.6$ | $22.2 \pm 2.8$ | $25.6 \pm 6.8$ |
| OpenVLA | $77.8 \pm 6.4$ | $45.8 \pm 4.2$ | $91.7 \pm 8.3$ | $55.6 \pm 2.8$ | $51.3 \pm 2.6$ | $27.8 \pm 2.8$ | $38.5 \pm 4.4$ |
| Ours | $\mathbf{92.6 \pm 3.7}$ | $\mathbf{100 \pm 0.0}$ | $\mathbf{95.8 \pm 4.2}$ | $\mathbf{94.4 \pm 2.7}$ | $\mathbf{94.9 \pm 5.1}$ | $\mathbf{63.6 \pm 5.2}$ | $\mathbf{59.5 \pm 2.4}$ |

| Method | pick toys then place to target | | | play basketball | push | | Average |
| --- | --- | --- | --- | --- | --- | --- | --- |
| | spiderman | penguin | pig | | push red button | push red then blue | |
| ATM | $5.3 \pm 1.3$ | $6.7 \pm 1.3$ | $5.3 \pm 3.5$ | $24.0 \pm 4.6$ | $4.0 \pm 2.3$ | $0.0 \pm 0.0$ | $6.4 \pm 2.2$ |
| LLARVA | $9.3 \pm 1.3$ | $9.3 \pm 1.3$ | $8.0 \pm 2.3$ | $10.7 \pm 3.5$ | $20.5 \pm 5.1$ | $2.8 \pm 2.8$ | $18.3 \pm 3.9$ |
| $\pi_0$-FAST | $16.0 \pm 2.3$ | $17.3 \pm 1.3$ | $9.3 \pm 2.7$ | $13.3 \pm 3.5$ | $20.5 \pm 2.6$ | $0.0 \pm 0.0$ | $21.2 \pm 3.0$ |
| OpenVLA | $2.7 \pm 1.3$ | $17.3 \pm 1.3$ | $2.7 \pm 2.7$ | $49.3 \pm 3.5$ | $23.1 \pm 4.4$ | $0.0 \pm 0.0$ | $37.2 \pm 3.4$ |
| Ours | $\mathbf{90.7 \pm 1.3}$ | $\mathbf{94.7 \pm 1.3}$ | $\mathbf{93.3 \pm 1.3}$ | $\mathbf{92.0 \pm 2.3}$ | $\mathbf{84.6 \pm 4.4}$ | $\mathbf{25.0 \pm 4.8}$ | $\mathbf{83.1 \pm 3.0}$ |

natural rotation: we examined a new task,"put knife" in RLBench and found our model struggles to succeed at this task. Interestingly, this simulated task features an unnatural whole-arm rotation to grasp the knife handle, based on the expert demonstrations created by the simulator's motion planning. As the movement between two key points is very long, we hypothesize that our model has difficulty learning this rotation. (2) Lack of Precision: we observed that ARM4R struggles with the 'screw bulb' task, a procedure requiring precise insertion into the bulb holder. In contrast, our model effectively performs other standard precision tasks, such as 'open drawer' which necessitate only standard precision.

### 4.3. Real Robot Evaluation

**Experimental Setup**. For our real experiments, we use a 7-DoF Kinova Gen3 robot mounted with a Robotiq 2F-85 adaptive gripper. We test our model and the baselines on 13 total tasks, grouped into five broad categories based on the dominant action: *pick*, *destack*, *stack*, *pick and place*, and *push*. For each task, training is performed using 190 episodes of every variation. Evaluation is conducted over 25 episodes per task, with results averaged across three different seeds to calculate the final success rate.

**Baselines**. We evaluate our model against four baselines in real-world settings: ATM (Wen et al., 2024), LLARVA (Niu et al., 2024), $\pi_0$-FAST (Pertsch et al., 2025) and Open-VLA (Kim et al., 2024). ATM utilizes a hierarchical framework to predict 2D point trajectories, which are then used to condition a policy, while LLARVA uses 2D point prediction as an auxiliary task for learning action prediction. In contrast, ARM4R predicts 3D point trajectories, a more intuitive and natural representation for robotic tasks. $\pi_0$-

FAST and OpenVLA are state-of-the-art VLA models pre-trained on the OpenX dataset, while ARM4R is trained on a significantly smaller dataset, with pre-training consisting exclusively of human video data. More details for these implementations are in Appendix D.1.

**Results**. Table 2 shows that ARM4R outperforms all baselines across all tasks, **achieving an average success rate of 83.1%**, compared to OpenVLA's 37.2% and ATM's 6.4%. ATM in particular does not perform well in our real setting despite training with a significantly larger number of demonstrations than we use in our fine-tuning. We believe that this significant gap in performance is due to how we track points: ARM4R utilizes 3D coordinates, while ATM relies on 2D. The use of 3D coordinates provides a more natural and accurate representation for robotic tasks, which may contribute to our model's improved performance.

In contrast to ATM, OpenVLA and $\pi_0$-FAST use a similar number of fine-tuning episodes to our evaluation setting. However, we believe that our superior performance over these baselines can again be attributed to our use of low-level 4D representations, which enable 3D scene understanding.

### 4.4. Ablation Studies

We conduct ablations to assess the importance of human video pre-training (Stage 1), and the robotic fine-tuning (Stage 2). All model versions in this section include robotic control fine-tuning (Stage 3). For this, we train the following versions: (i) Stages 1, 2, and 3; (ii) Stages 1 and 3; (iii) Stages 2 and 3, and (iv) Stage 3 only.

**Human Video Pre-Training**. Figure 3 shows that the model with all stages performs better on all tasks than the Stages 2+3 model, indicating that pre-training on the human dataset provides a large benefit compared to only training

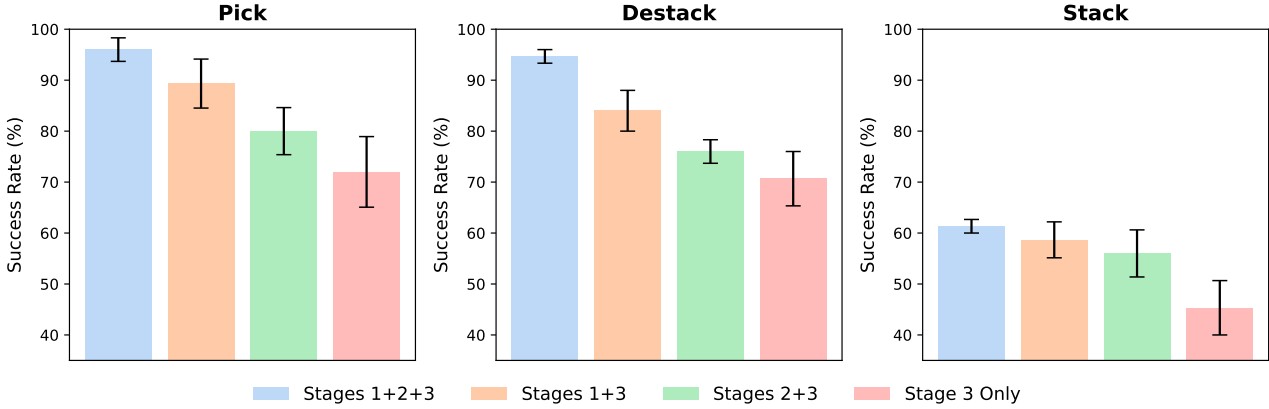

Figure 3: **Ablation Study for Stages 1 and 2.** We train ARM4R on three real tasks in the Kinova setting, ablating Stages 1 and 2. The results indicate that while both stages improve performance, Stage 1 has a more significant impact.

Table 3: **Pre-training approaches comparison.** We compare ARM4R to several other robotic models that leverage pre-training on three tasks with a Kinova robot. We find that our approach yields the best average success rate.

| Method | pick cube | stack cubes | destack cubes |
|---|---|---|---|
| MVP | 75.00 | 18.75 | 81.25 |
| RPT | 87.50 | 31.25 | 93.75 |
| Octo | 56.25 | 12.50 | 37.50 |
| ATM | 7.11 | 2.00 | 6.67 |
| OpenVLA | 68.75 | 31.25 | 53.33 |
| LLARVA | 93.75 | 56.25 | **100.00** |
| ARM4R | **96.0 ± 2.3** | **61.3 ± 1.3** | 94.7 ± 1.3 |

for 3D point tracking on robotics videos. The performance boost observed when adding Stage 1 to Stage 3 is greater than the boost from adding Stage 2 to Stage 3, indicating that 4D pre-training on human videos provides a larger increase in performance than robotic videos. The key resulting insight is that when sufficient robotic pre-training data is unavailable, human video data can be a viable alternative, provided the proper 4D representations are used.

**Robotic Video Fine-Tuning**. The ablation results shown in Figure 3 reveal that adding robotic video fine-tuning (Stages 2+3; green) leads to improved performance over models trained solely for robotic control (Stage 3; pink). Adding Stage 2 to the training regime still improves performance, as the model performing all stages (blue) yields the highest success rate. As mentioned in Section 3.3, Stage 2 is useful in addressing the distribution shift and embodiment gap when switching from human to robotic data.

### 4.5. Additional Experiments

We conduct additional experiments to assess the benefits of pre-training, generalization of 3D point representations, and

robustness to noise. More experiments are in Appendix A.

**The Effectiveness of Pre-training**. In order to study the effectiveness of pre-training on the 3D point track prediction task, we take three tasks from our real setting: *pick cube*, *destack cubes*, and *stack cubes*, and compare to other works that use pre-training. MVP focuses on pre-training the vision encoder using human data, while RPT focuses on pre-training with visual and proprioceptive states. Octo, which is a transformer-based policy, is pre-trained on the OpenX dataset, similar to the VLA models LLARVA and OpenVLA. Lastly, ATM pre-trains a 2D point track transformer whose output is used to condition a policy.

The results are shown in Table 3. It can be seen that our pre-training improves performance over the baselines. ARM4R outperforms other representation learning based pre-training methods, such as MVP, RPT, Octo and ATM, validating the benefits of using a 4D point-tracking based representation. In addition, while the two VLA baselines (OpenVLA and LLARVA) perform well, our approach still surpasses their results, possibly demonstrating the importance of using low-level representations as opposed to language decoders that were pre-trained on high-level vision-language tasks.

Table 4: **Success rate (%) of ARM4R on cross-robot setting.** We fine-tune the pre-trained model for motor control on different robots and report success rates of cube tasks.

| Pre-train | FT | Robot | pick | stack | destack |
|---|---|---|---|---|---|
| Epic | Kinova | Kinova | 96.0 ± 2.3 | 61.3 ± 1.3 | 94.7 ± 1.3 |
| – | – | Franka | 73.3 ± 2.7 | 49.3 ± 5.8 | 65.3 ± 3.5 |
| Epic | Kinova | Franka | 93.3 ± 1.3 | 56.0 ± 2.3 | 97.3 ± 1.3 |
| – | Kinova | Franka | 81.3 ± 1.3 | 52.0 ± 2.3 | 73.3 ± 2.7 |

**Generalization from Kinova to Franka**. In order to study how our low-level 4D representations can help a model

generalize across different robots, we perform an ablation experiment involving fine-tuning ARM4R on Kinova robot videos, and fine-tuning for control on a 7 DoF Franka Emika Panda robot. We note that besides having different robots, the two setups also have very different configurations, as the Kinova robot is mounted on a stand as part of a bimanual setup, while the Franka robot is mounted on a table.

Despite these significant differences, the results in Table 4 show that adding the human video pre-training and Kinova video fine-tuning improves the average performance on the Franka robot by 19.6%. This supports our claim that the 4D representations are sufficiently generalizable to transfer across different robotic setups.

**Robustness Analysis**. We conduct an experiment to test how ARM4R handles dynamic changes in its environment. We test the model on three cube tasks (Pick, Stack, and Destack) in our real Kinova setting, manually moving the cube during rollout when the arm reaches 1/3 and 2/3 of its descent. The results are presented in Table 5, and show consistent performance despite movement of the target object. This highlights the model's robustness to environmental and object-level changes, which we believe is partly due to the fact that it predicts each step in a trajectory and thus interacts with the environment continuously.

Table 5: **Performance in dynamic environments.** The cube is disturbed at varying phases of the robot trajectory.

| Disturbance Time | Pick | Stack | Destack |
|---|---|---|---|
| No Disturbance | 96.0±2.3 | 61.3±1.3 | 94.7±1.3 |
| At 1/3 Descent | 94.7±3.5 | 60.0±0.0 | 93.3±1.3 |
| At 2/3 Descent | 92.0±2.3 | 57.3±1.3 | 90.7±1.3 |

To further assess ARM4R' robustness in dealing with factors like camera noise, occlusions, and sensor drift, we perform additional evaluations on the three cube tasks under the following conditions: (1) Dim lighting: ambient light is reduced to 50%, (2) Background distractors: dynamic background changes, such as people walking by or movement of background curtains, and (3) Tabletop distractors: two random objects are placed near the target. The results are presented in Table 6. Overall, ARM4R demonstrates robustness to lighting and background changes. This is likely due to the use of attention pooling, which guides the model to focus on the target region rather than background features. This hypothesis is supported by the observed performance drop when tabletop distractors are introduced.

## 5. Conclusion

In this work, we demonstrate that our pre-training approach from human video data to robot learning is effective in addressing longstanding challenges of robotic learning pre-training. We introduced ARM4R, an Auto-regressive

Table 6: **Performance under different real-world robustness conditions.**

| Noise Source | Pick | Stack | Destack |
|---|---|---|---|
| Standard | 96.0±2.3 | 61.3±1.3 | 94.7±1.3 |
| Dim Light | 86.7±1.3 | 52.0±2.3 | 81.3±2.7 |
| Background Distractor | 94.7±1.3 | 57.3±1.3 | 90.7±1.3 |
| Table-top Distractors | 81.3±1.3 | 48.0±2.3 | 74.7±1.3 |

Robotic Model that leverages low-level 4D representations by lifting 2D representations into 3D using monocular depth estimators, and tracking 3D points in videos. Our results in simulation and real-world setups show that our method consistently outperforms existing methods across diverse robotic tasks, showcasing its superior transferability and generalization capabilities. More broadly, our approach shows that training solely on human video data can lead to better performance than methods like OpenVLA that are pre-trained on robotic data alone. This suggests that effective pre-training can be achieved without the need for large-scale robotic datasets by bridging the gap between human-centric visual data and robotic applications, unlocking new possibilities for scalable and data-efficient robotics. As we continue to explore the boundaries of representation learning for robotics, ARM4R lays a foundation for future research into autonomous systems that can learn from the vast repository of human experience available in video data.

## 6. Limitations and Future Work

While ARM4R offers substantial benefits for pre-training with human video data for robotic learning, it is important to recognize certain limitations that accompany our approach. Our approach tracks 3D points in camera coordinates, leading to learned representations that combine object and camera motion. This coupling makes it difficult for the model to disentangle the two, leading to potentially inaccurate predictions due to a lack of invariance to camera intrinsics and motion. An improvement addressing this concern could involve pre-training on 3D tracks in world coordinates, leveraging recent dynamic SLAM methods such as MonST3R (Zhang et al., 2024) or MegaSAM (Li et al., 2024b), which we leave for future work. Other improvements could include scaling the pre-training data to help the model generalize better to different camera viewpoints, or using multi-view fusion to reduce the dependency on a single viewpoint to improve robustness to occlusions. Another line of future work could focus on selectively tracking only relevant or moving points instead of a fixed uniform grid across frames. This would allow greater resolution in areas with small objects, and also help the model focus on objects critical to the task, improving its ability to disentangle object motion from background noise and camera movement.

## Acknowledgments

We would like to thank Abrar Anwar, Chancharik Mitra, and Yida Yin for their helpful feedback and discussions, as well as Rodolfo Corona for his assistance with data collection. We also thank Nicole Walters for creating our lovely logo. This project was supported in part by BAIR's industrial alliance programs.

## Impact Statement

In this paper, we demonstrate how pre-training on low-level 4D representations from human video data can benefit robotic action prediction. This suggests that effective pre-training can be achieved without the need for large-scale robotic datasets by bridging the gap between human-centric visual data and robotic applications, unlocking new possibilities for scalable and data-efficient robotics. Finally, we note that this paper presents work whose goal is to advance the field of Machine Learning. There are many potential societal consequences of our work, none which we feel must be specifically highlighted here.

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

# Appendix

Here, we provide qualitative results of our 3D point tracking for both in-domain and out-of-domain videos (Appendix A), statistics of the used datasets (Appendix B), implementation details (Appendix C), evaluation details (Appendix D), and robotic setup details (Appendices E and F).

## A. Additional Experiment Results

### A.1. Qualitative Results of 3D Points Tracking.

We conduct additional experiments to evaluate our pre-trained model's ability to track 3D points. Specifically, we run inference on a few randomly chosen episodes from Epic-Kitchens100 (Damen et al., 2018) (in-domain human videos), Ego4D (Grauman et al., 2021) (out-of-domain human videos), Kinova robot videos (in-domain robot videos) and Open X Embodiment (Collaboration et al., 2023) (out-of-domain robot videos).

In Figure 4, we present the tracking results on human videos from a version of our model that has undergone human video pre-training (Stage 1). The top two rows show the results for an episode from Epic-Kitchens (in-domain human videos) with the action "stir potatoes." The bottom two rows display monocular human videos and their corresponding 3D point tracking predictions for an episode from Ego-4D (out-of-domain human videos) with the action "pick up plate".

In Figure 5, we present the tracking results on robot videos, from a version of our model that has undergone human video pre-training on Epic-Kitchens100 as well as robot video fine-tuning on Kinova demonstration videos (Stage 1+2). The top four rows display monocular robot videos and their corresponding 3D point tracking predictions for two episodes from in-domain Kinova robot videos, with the actions "push red button" and "place spiderman into bowl" respectively. The bottom two rows show the results for an episode from the Autolab subset of the OpenX Embodiment dataset (out-of-domain robot videos) with the action "pick the tiger and place it into bowl."

These visualizations verify that our model is not overfit to a certain dataset or robotic setup, but can in fact generalize well to new videos.

## B. Additional Dataset Details

### B.1. Epic-Kitchens100

*Epic-Kitchens100* (Damen et al., 2018) is a large-scale, egocentric video dataset designed for action recognition and understanding in daily kitchen activities. Captured from a first-person perspective using head-mounted cameras, the dataset provides rich, untrimmed video recordings of individuals performing various cooking and kitchen-related

tasks. It features a diverse range of object interactions, fine-grained action labels, and naturalistic, unscripted activities, making it particularly valuable for studying human-object interactions and long-term temporal dependencies.

The dataset includes diverse hand-object interactions, described by combinations of 97 verbs (for the hand motions) with 300 nouns (for the object categories). In our human video pre-training stage, we use almost all the labeled episodes available in the original dataset. We first subsample videos at 10 *fps*, an experimentally chosen rate, as the original 50 *fps* provides unnecessary redundancy for slow movements. We then model the duration distribution of all 75,886 episodes, and filter out $\approx 1\%$ of episodes that are of length $> 256$ frames. As a result, we get a final set of 75,041 episodes for pre-training. For each episode, we use a simple 'verb + noun' instruction derived from the official annotation files.

### B.2. RLBench Robot Episodes

RLBench (James et al., 2020) is a large-scale benchmark dataset for robotic learning, designed to facilitate research in vision-based reinforcement learning and imitation learning. It consists of a diverse set of robot manipulation tasks performed in a simulated environment using a Franka Emika Panda arm. The dataset provides high-quality demonstrations with multi-modal observations, including RGB images, depth maps, and proprioceptive data.

In our experiments, we use $128 \times 128$ resolution images for training. For most tasks, we use the 'front rgb' and 'wrist rgb' views for point track and control fine-tuning. However, in some cases, we find that using other views yields better performance (details on task-specific implementations are provided in Appendix C). For robot control, we use end-effector control: $\mathbf{x} = (x, y, z, \theta_x, \theta_y, \theta_z)$, where $(x, y, z)$ is the position and $(\theta_x, \theta_y, \theta_z)$ the Euler angles for orientation. We also have a one-dimensional binary element to control the gripper. For language instructions, we use variation 0 from the official list of instructions for all tasks. We do not subsample episodes for our RLBench experiments.

## C. Additional Implementation Details

### C.1. Architecture of Auto-regressive Model

Here, we provide details on processing the visual input of the auto-regressive model to support two views when adapting to robot control fine-tuning. The images from both views are fed separately into the image encoder to obtain the image embeddings $z_t^{im}$ for each view. Each view is then pooled using attention pooling with the state embeddings to form the image tokens $z_t^{obs}$. Next, we project each token to half of its original hidden dimension ($768 \rightarrow 384$ in our implementation) and concatenate them to obtain the final

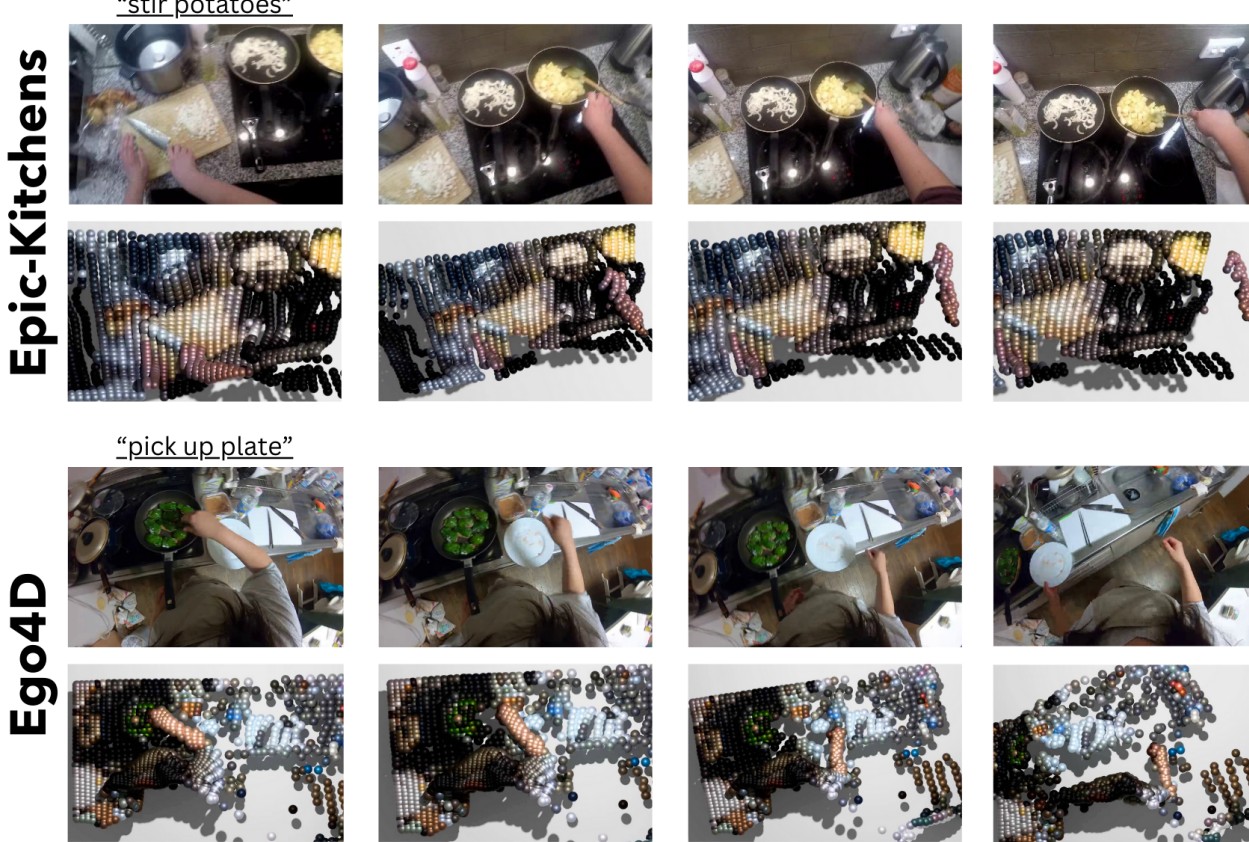

Figure 4: Visualization of ARM4R's 3D Point Track results on randomly chosen Epic-Kitchens (in-domain) and Ego-4D (out-of-domain) human videos.

image tokens, incorporating information from both views.

### C.2. Training Recipes

We used the following hyperparameters for the three stages of training:

Table 7: Training Hyperparameters for the three stages.

| Hyperparameter | Stage 1 | Stage 2 | Stage 3 |
|---|---|---|---|
| Learning Rate | $5 \times 10^{-4}$ | $5 \times 10^{-4}$ | $5 \times 10^{-3}$ |
| Weight Decay | $1 \times 10^{-2}$ | $1 \times 10^{-2}$ | $1 \times 10^{-2}$ |
| Batch Size | 256 | 256 | 256 |
| Number of Epochs | 5 | 20 | 10-50 |

We note that for Stage 3, we trained our model for a variable number of epochs depending on the downstream task, until the loss converged.

## D. Simulation on RLBench

We evaluate our model on 12 tasks in RLBench for our simulation setup. Each task includes multiple variations, and we generate 190 episodes using their data generation

script for ARM4R training. In most cases, we follow the task setup of PerAct (Shridhar et al., 2023) and use the 'front rgb' and 'wrist rgb' views. We use $C = 16$ ($C$ is the context window of the auto-regressive model). The detailed task-level configuration is provided below.

**Open Drawer**. The task is to open one of three drawers. The success metric is a full extension of the prismatic joint of the target drawer. We use the 'front rgb' and 'wrist rgb' views. The context window of the model is $C = 16$.

**Meat off Grill**. The task is to take either a piece of chicken or steak off the grill and put it on the side. The success metric is the placement of the specified meat on the side, away from the grill. We use the 'front rgb' and 'wrist rgb' views. The context window of the model is $C = 16$.

**Turn Tap**. The task is to turn either the left or right handle of the tap. Left and right are defined according to the orientation of the faucet. The success metric is the joint of the specified handle being at least $90°$ away from the starting position. We use the 'front rgb' and 'wrist rgb' views. The context window of the model is $C = 16$.

**Put Money**. The task is to pick up the stack of money and

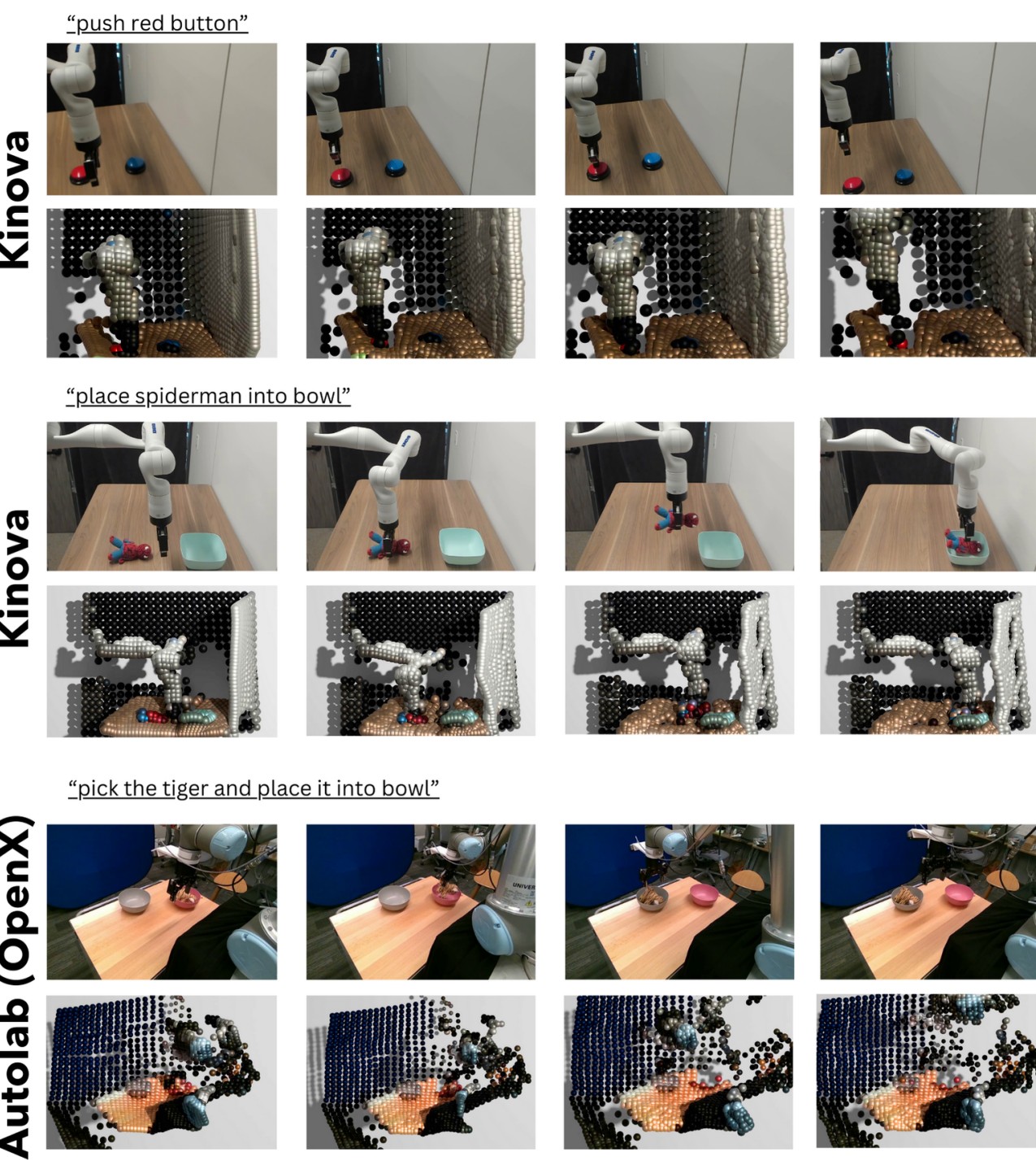

Figure 5: Visualization of ARM4R's 3D Point Track results on randomly chosen Kinova (in-domain) and Open X-Embodiment (out-of-domain) robot videos.

place it on the specified shelf of a safe. The safe has three shelves: top, middle, and bottom. The success metric is the placement of the stack of money on the specified shelf in the safe. We use the 'front rgb' and 'overhead rgb' views.

The context window of the model is $C = 16$.

**Push Buttons**. The task is to push the colored buttons in the specified sequence. There are always three buttons present in the scene, whose colors are sampled from 20 options, and

the number of buttons to press is between one and three. The success metric is all specified buttons being pressed in the right order. We use the 'front rgb' and 'wrist rgb' views. The context window of the model is $C = 32$.

**Sweep Dustpan**. The task is to sweep the dirt particles into the specified dustpan. There are two dustpans, one short and one tall, and both are always present in the scene. The success metric is all five dirt particles being inside the specified dustpan. We modified this task by adding a variation with a different-sized dustpan. We use the 'front rgb' view only, repeated twice, for this task. The context window of the model is $C = 16$.

**Slide Block**. In this task there is a block and four colored squares in the scene (green, blue, pink, and yellow). The task is to slide the block onto either the green or pink squares. The success metric used is some part of the block being on the specified target square. The original task only had one target square, and we modified it by adding three additional colored squares — one target and two distractors. We use the 'front rgb' view only, repeated twice, for this task. The context window of the model is $C = 16$.

**Close Jar**. The task is to screw in the lid on the jar with the specified color. There are always two colored jars in the scene, one target jar and one distractor jar. The success metric used is the lid being on top of the specified jar and the robot gripper not grasping any object. We modified this task so that the target jar color is drawn from a list of three possible colors (red, maroon, and lime ). The color for the distractor jar was still chosen out of 20 options. We use the 'front rgb' and 'wrist rgb' views. The context window of the model is $C = 32$.

**Screw Bulb**. There are two bulb holders of different colors, and the task is to pick up a light bulb from the stand specified by color and screw it into the bulb stand. The color of the target holder is sampled from two colors, while the color of the distractor holder is sampled from the original 20 color options. The success metric used is the bulb from the specified holder being inside the bulb stand. We modified this task to use three colors for the target holder (yellow, purple and silver) rather than 20 as in the original task specification. We use the 'front rgb' and 'wrist rgb' views. The context window of the model is $C = 16$.

**Place Wine**. The task is to pick up the wine bottle and place it at the specified location in a wooden rack. The rack has three locations: left, middle, and right. The success metric is the placement of the bottle on the specified location in the rack. We use the 'front rgb' and 'wrist rgb' views. The context window of the model is $C = 16$.

**Reach and Drag**. The environment has a cube, a stick, and four possible colored target squares. The task is to pick up the stick and use it to drag the cube to the target square of

a specified color. The other three squares are considered distractors. The success metric used is some part of the block being inside the target's area. We modified this task to sample the target color from a list of three colors (maroon, magenta, teal). The colors for distractor squares are still sampled from 20 options. We use the 'front rgb' and 'wrist rgb' views. The context window of the model is $C = 16$.

**Stack Blocks** . The scene starts with 8 blocks and a green platform. Four of the blocks are of a target color, and the other four have a distractor color. The task is to stack $N$ blocks of the target color on the green platform. The success metric is $N$ blocks being inside the area of the green platform. We use the 'front rgb' and 'wrist rgb' views. The context window of the model is $C = 16$.

### D.1. Baselines in Real Experiments

**ATM**. We reproduce ATM (Wen et al., 2024) as a baseline for our Kinova real-world experiment setup, following the provided code and instructions [2]. In the first stage, we use all Kinova robot episodes (5 tasks, each with 200 episodes per variation) to train a track transformer using ground truth point tracks generated by Co-Tracker (Karaev et al., 2025). In the second stage, we take the best checkpoint of the track transformer to train a policy for each task separately, consistent with ARM4R's real-world setup. ATM uses a 7-dimensional joint pose and one-dimensional gripper state in its policy. To adapt it to our data format, we modify the implementation to end-effector control, predicting a 3-dimensional $(x, y, z)$ position, a 4-dimensional quaternion rotation, and a 1-dimensional gripper state.

**OpenVLA**. We also test OpenVLA (Kim et al., 2024) on our Kinova robot setup, following their fine-tuning code and instructions [3]. We fine-tune OpenVLA using LoRA (Hu et al., 2021), with rank 32 and a batch size of 16, training until convergence. To adapt OpenVLA to our control setting, we convert our absolute proprioceptive states to 3-dimensional delta position and 3-dimensional delta rotation (Euler angles), with an additional binary gripper dimension. We subsample our original collected data at a ratio of 10, since the difference between consecutive steps in the original data is too small for delta control, given the accuracy limit of OpenVLA ($10^{-3}$).

**LLARVA**. We reproduce LLARVA (Niu et al., 2024) as a baseline for our real-world Kinova experiment setup, following the provided code and instructions [4]. We fine-tune their released pre-trained checkpoint on our Kinova demonstrations for single-task policies (5 tasks, each with 200 episodes per variation). We use their default learning rate

---

[2] https://github.com/Large-Trajectory-Model/ATM
[3] https://github.com/openvla/openvla
[4] https://github.com/Dantong88/LLARVA

and batch size, and train for 4 epochs for each task.

$\pi_0$**-FAST**. We fine-tune $\pi_0$-FAST as a baseline for our Kinova real-world experiment setup, following the provided code and instructions[5]. We use their released base autoregressive $\pi_0$-FAST model for fine-tuning a single-task policy (200 demonstrations per variation). We use joint position control instead of joint velocity control, fine-tuning the pretrained model to predict the absolute 7-DoF joint pose plus 1-DoF gripper status. We follow the default learning rate used in their implementation.

# E. Real-World Kinova Experiments

## E.1. Hardware

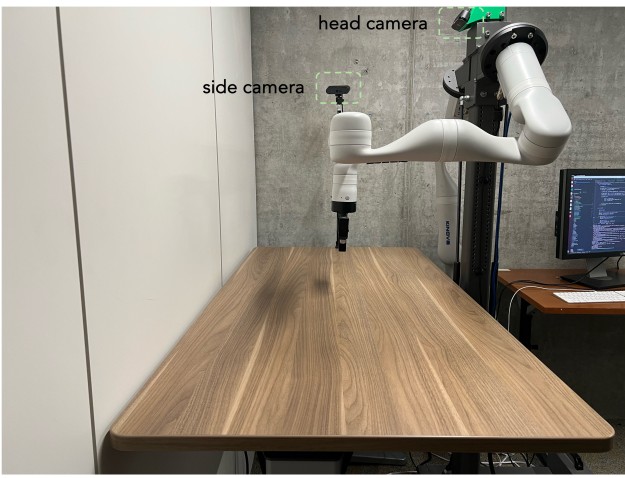

Figure 6: The real-world experiment setup of Kinova robot.

For our primary real-world experiments, we use a Kinova Gen3 7 DoF manipulator with a Robotiq 2F-85 gripper, as shown in Figure 6. It is mounted on a base that mimics the human shoulder orientation and height.

We set up two cameras (Logitech BRIO 4K camera) to observe the table-top manipulation scene. One is mounted at an ego-centric pose, and the other is mounted on the side of the table.

We use the MoveIt motion planning framework for inverse kinematics and end-effector position control. It takes the end effector position objective from the model, and executes linear trajectories in the Cartesian space.

## E.2. Data Collection

To collect task demonstrations, we develop an automated data collection procedure to record episodes of these demonstrations. In this procedure, we give the ground truth locations of all objects on the table, and procedurally generate

task objectives, demonstrations, and accompanying task instruction labels. Domain randomization is applied to diversify robot home position, grasping approach trajectory, and target pose.

## E.3. Task Building

We build 5 tasks under the Kinova real robot setup. The configuration of each task and its variations are shown in Figure 7. The details of each task are described as follows.

**Pick Cube**. The episode starts with the arm in the home position. The robot moves to pick up the cube, placed at a random location on the table within the manipulator's workspace. The episode recording stops after the robot picks up the object and moves up by a certain distance.

**Stack Cubes**. The episode starts with the arm in the home position. After picking up the cube from point A as in the pick cube task, the robot stacks it on another object already present in the scene at point B according to the instruction. The episode recording stops after the robot releases the object and moves up by a certain distance.

**Destack Cubes**. The episode starts with the arm in the home position. After picking up the top cube from a stacked pair at point A as in the grasping task, the robot moves the grasped cube to another location, point B. The episode recording stops after the robot releases the object and moves up by a certain distance.

**Pick & Place Toys/Basketball**. The episode starts with the arm in the home position. After picking up the toy described in the instruction from point A, the robot moves and places the toy into a bowl or basket at point B. The episode recording stops after the robot releases the object and moves up by a certain distance.

**Push Buttons**. The episode starts with the arm in the home position. Following the instruction, the arm moves to a specific height above the assigned button at point A, closes the gripper, pushes the button, then moves to push another button at point B. The episode recording stops after the robot releases the object and moves up by a certain distance.

# F. Real-World Franka Experiments

## F.1. Hardware

We use a Franka Emika Panda robot with a Franka gripper for real robot data collection and evaluations. The Logitech BRIO 4K cameras positioned to the left and right of the Franka robot provides double-view RGB (without depth data) vision input to our model, as shown in Figure 8. Camera autofocus is disabled, and the data is captured at 640x480 resolution.

---

[5]https://github.com/Physical-Intelligence/openpi

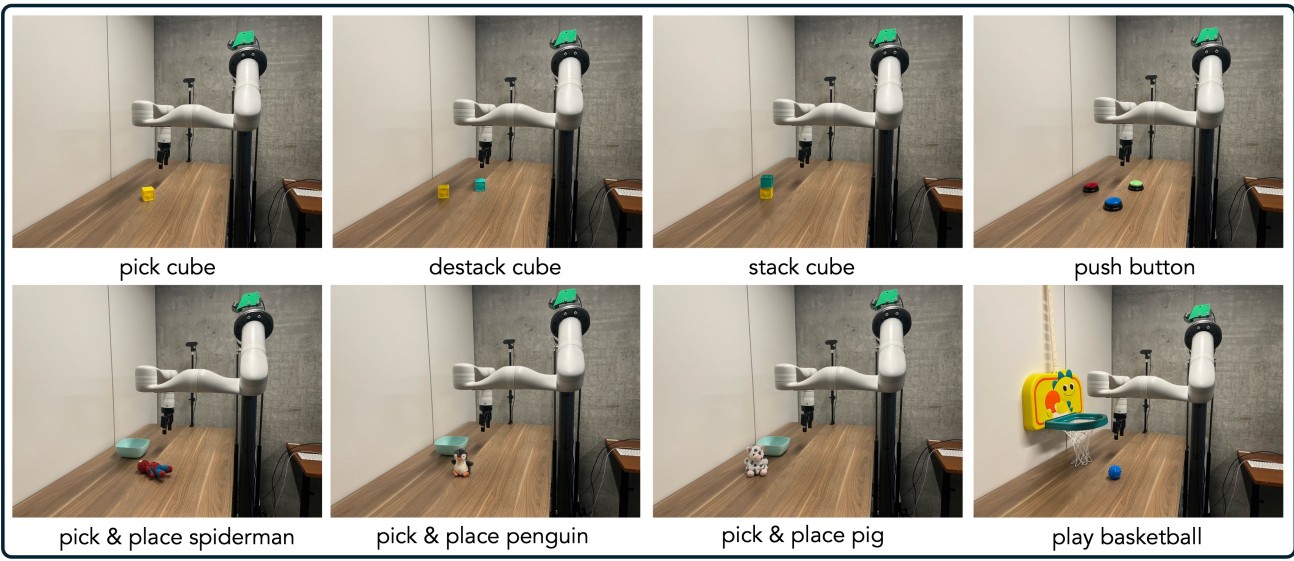

Figure 7: Task building of real-world Kinova setup.

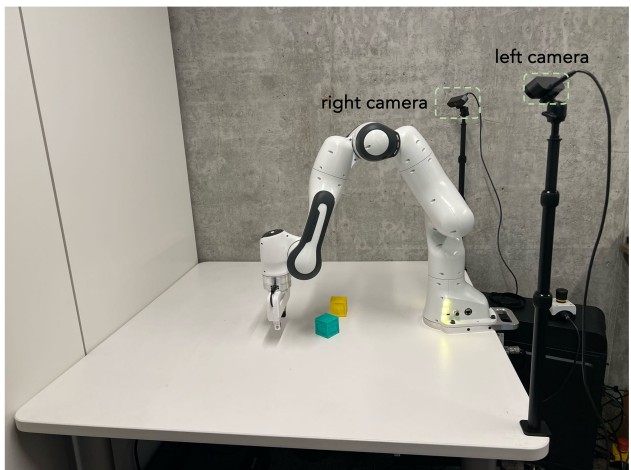

Figure 8: The real-world experiment setup of Franka robot.

in Figure 8. The details of each task are described as follows.

**Pick Cube**. The episode starts with the arm in the home position. The robot moves to pick up the cube from point A. The episode recording stops after the robot picks up the object and moves up by a certain distance.

**Stack Cubes**. The episode starts with the arm in the home position. After picking up the cube from point A as in the grasping task, the robot stacks it on another object already present in the scene at point B according to the instruction. The episode recording stops after the robot releases the object and moves up by a certain distance.

**Destack Cubes**. The episode starts with the arm in the home position. After picking up the top cube from a stacked pair at point A as in the grasping task, the robot moves the grasped cube to another location, point B. The episode recording stops after the robot releases the object and moves up by a certain distance.

### F.2. Data Collection

We use the data collection code and process from https://github.com/Max-Fu/franka-scripted to collect data for automated tasks. The script generates data for an arbitrary number of episodes. For each episode, the process generates x-y positions on the table plane using a uniform random distribution for each axis. The script directs the robot to place the object at each location and then collects the camera and joint information as the robot is moving.

### F.3. Task Building

We build cube tasks under the Franka real robot setup. The configurations of each task and its variations are shown

