# OpenReview forum: "Pre-training Auto-regressive Robotic Models with 4D Representations"
_ICML.cc/2025/Conference — ICML 2025 poster_

### Official Review · Reviewer_kSQ4 · 2025-03-09

**Overall Recommendation:** 3

**Summary:**

The paper presents ARM4R, an Auto-regressive Robotic Model that learns 4D representations from human videos to enhance robotic learning.
- 4D Representation Learning: Transforms 2D videos into 3D point tracks, enabling temporal and spatial understanding.
- Efficient Transfer to Robotics: Pre-training on human videos outperforms training directly on robotic datasets.
- Improved Performance: ARM4R surpasses baselines like PerAct, OpenVLA, and ATM in both simulated and real-world tasks.
- Cross-Robot Generalization: Demonstrates strong generalization across different robotic platforms (e.g., Kinova and Franka).

## update after rebuttal
Thanks for the rebuttal. All my concerns are resolved. I maintain my initial rating.

**Claims And Evidence:**

The paper makes strong claims about transfer learning in robotics using 4D representations, supported by empirical evidence, but some potential concerns remain:
- Causal analysis: The causal relationship between 4D representations and robotic success is not thoroughly analyzed, leaving room for further investigation.
- Optimality of 4D Representations: The paper lacks formal proof that 4D representations are the optimal choice for robotic learning. It does not compare 4D representations to alternative 3D motion-based methods, leaving this claim unverified.
- Geometric Alignment: The assertion that linear transformation preserves geometric alignment between human and robotic states is not rigorously justified, raising questions about its theoretical validity.
- 2D-to-3D Lifting: The approach of lifting 2D video data into 3D using monocular depth estimation is not thoroughly justified, leaving uncertainty about its robustness and generalizability.

**Essential References Not Discussed:**

No.

**Experimental Designs Or Analyses:**

- Performance Discrepancies Across Tasks: The model achieves high overall success rates, but specific tasks reveal performance inconsistencies. Rotational motion tasks (e.g., "screw bulb") show worse performance compared to the voxel-based model PerAct, while planar force-based tasks underperform relative to the LLM-based model LLARVA, which also uses intermediate 2D states as guidance. A deeper analysis of why these specific failure cases occur could lead to future improvements in representation learning.
- Real Robot Evaluation: For pre-training models, 3-task subset may not unintentionally favor ARM4R performing best.
- Scalability and Data Efficiency: There is no discussion on whether smaller human datasets (e.g., 5K or 10K videos) would still yield strong performance.
- Potential Dataset Bias: The model is pre-trained on Epic-Kitchens100, which focuses on household activities, which raises concerns about bias.

**Methods And Evaluation Criteria:**

- Failure Case Analysis: The study does not evaluate failure cases, such as camera noise, occlusions, or sensor drift, which are critical for assessing real-world robustness.
- Generalization to Dynamic Environments: While experiments are conducted in controlled settings, their applicability to highly dynamic or unpredictable real-world environments remains unverified.

**Other Comments Or Suggestions:**

Null

**Other Strengths And Weaknesses:**

Null

**Questions For Authors:**

See the sections above for detailed feedback.

**Relation To Broader Scientific Literature:**

Good.

**Theoretical Claims:**

Yes.

---

> ### Author Rebuttal · Authors · 2025-04-01
>
> We thank the reviewer for the insightful comments. We address the concerns raised below:
>
> 1. **Causal Analysis & Optimality of 4D Representations:** As stated in the paper, one of the main benefits of using 4D representations is that the 3D point tracks in a robotic setting are described by linear transformations constructed using the robot states. We present a simple proof of this below:
>
> Consider a $n$-DoF open-chain manipulator, initially in its reference configuration $\theta_1=0, \cdots, \theta_n=0$. Suppose the manipulator is commanded to a new position $\theta_1=\theta_1^d, \cdots, \theta_n=\theta_n^d$. Next, we take any arbitrary point $p$ on the body of the manipulator lying between joints $i, i+1$. Let $p$ be described by the transformation $g_{i,p} (0)$ in joint $i$'s frame. Then, $p$'s new position in the base frame can be described by:
> $$
> g_{1,p} (\theta) = e^{{\hat{\xi_1}}\theta_1^d}\cdots e^{{\hat{\xi_i}}\theta_i^d} g_{i,p} (0)
> $$
>
> where ${\hat{\xi_j}}$ are the twists associated with each joint. Since this product of 4x4 SE(3) matrices represents a linear transformation, points on the robot body as well as attached rigid bodies evolve through linear transformations in terms of the robot states.
>
> Overall, this proof shows that learning 3D point track prediction is related to and can benefit robotic control, as validated by our ablation experiments. Additionally, for overactuated manipulators (7 DoF), the robot learning problem simplifies to tracking arbitrary trajectories, as configuration space limitations are less restrictive. This allows the model to benefit more effectively from the 3D point tracking pre-training task.
>
> 2. **Comparison to 2D and 3D methods:** We note that in our paper (see Tables 1/2), we compare ARM4R with both 2D and 3D approaches. ARM4R outperforms 2D motion-based methods (e.g., LLARVA, ATM) and 3D-based methods (e.g., ManiGaussian). We also include an additional comparison below with a recent 3D method, 3D-VLA, further demonstrating the strength of 4D representations for robotic learning.
>
> |Method|put knife|pick up cup|take umbrella|Average|
> |----|--|------|---|-------------|
> |Image-BC ViT|0|4|0|1.33 |
> |C2FARM-BC|4|28|48|26.67|
> |LLARVA|4|52|32| 29.33|
> |PerAct|20|56|76|50.67|
> |3D-VLA|68|40|52|53.33|
> |**ARM4R**|28.8±1.50|99.2±0.80|82.4±2.03|70.13±1.53|
>
> 3. **Geometric Alignment:** We clarify that our claim regarding geometric alignment refers not to alignment between human and robotic states, but between 3D point tracks in a robotic setting and the robot states. As shown in the proof above, these point tracks are described by linear transformations of the robotic states, which suggests that 4D representations are a natural choice for robotic control. We will clarify this in the final version.
>
> 4. **2D-to-3D Lifting:**  An alternative to monocular depth estimation is to use multiview cues and triangulation, which is more appropriate for dynamic camera setups. However, since our deployment in both real-world and simulation environments involves static cameras, monocular depth estimation is more suitable. This is the common and simple practice used in many 3D tracking papers, such as SpatialTracker and SoTA method DELTA.
>
> 5. **Scalability:** As the reviewer suggested, we add a scaling experiment, where we change the number of demonstrations used in our Epic-Kitchens pre-training. Specifically, we decrease the number of human videos in the pretraining to ⅓ and ⅔ of the original dataset used. We train the model for all three stages, and evaluate the performance on the three cube tasks. The results are summarized below:
>
> |Portion of the Epic Videos in Pre-Training|pick|stack|destack|
> |:------------------------------------------:|:------------:|:------------:|:------------:|
> |1/3|86.7$\pm$1.3|56$\pm$2.3|82.7$\pm$3.5|
> |2/3|90.7$\pm$1.3|58.7$\pm$2.7|89.3$\pm$2.7|
> |1|96.0$\pm$2.3|61.3$\pm$1.3|94.7$\pm$1.3|
>
> The results show that as the size of the pre-training set increases, the performance also tends to increase.
>
> 6. **Real Robot Evaluation:** We note that the 3 tasks of pick, stack and destack in Table 3 are commonly used for evaluation in several baselines, including LLARVA, Octo, RPT, and MVP. The main reason we follow the same procedure in this experiment is that comparing ARM4R to 6 different pre-training approaches on all 13 tasks would require a significant amount of demonstrations, as every baseline would need to be fine tuned for a fair comparison. Specifically, fair fine-tuning for each baseline would demand approximately four times the data currently used. We plan to expand the evaluation to include additional tasks in the final version.
>
> 7. **Monocular Depth Estimation:** See point 2 under Reviewer Zx5c.
>
> 8. **Robustness Analysis:** See point 3 under Reviewer vJYs.
>
> 9. **Failure Case Analysis:** See point 1 under Reviewer vJYs.
>
> 10. **Generalization to Dynamic environments:** See point 2 under Reviewer vJYs.
>
> 11. **Dataset Bias:** See point 4 under Reviewer vJYs.

---

### Official Review · Reviewer_vJYs · 2025-03-11

**Overall Recommendation:** 3

**Summary:**

The paper presents the ARM4R (Auto-regressive Robotic Model 4R), a novel approach to improve robotic pre-training by utilizing low-level 4D representations derived from human video data. The paper identify a significant challenge in the existing robotic models, which stems from a lack of large-scale, diverse datasets compared to those available in natural language and computer vision.

ARM4R addresses this gap by implementing a method that involves 3D point tracking, transforming 2D video representations into a format suitable for capturing essential physical world properties necessary for robotic manipulation. The main findings of the paper indicate that pre-training solely on human video data can yield superior performance compared to models trained exclusively on robotic data. Specifically, ARM4R surpasses established baselines, such as PerAct and OpenVLA, across multiple tasks in simulated and real environments with a 7-DoF Kinova Gen3 robot.

Key contributions include:

1. A novel pre-training approach that enhances understanding of the physical world through the utilization of low-level 4D representations from unlabeled human videos.
2. Evidence that training on human video data can lead to improved performance in robotic tasks, supporting the idea of efficient transfer learning from human actions to robotic control scenarios.
3. The model's capability to retain a shared geometric structure between 3D points and robot states, which facilitates better generalization across various robotic tasks and environments.

**Claims And Evidence:**

The claims made in the paper regarding ARM4R's effectiveness in enhancing robotic pre-training are largely supported by specific performance metrics and comparisons with other models. Here are the key points of evidence that reinforce the claims, along with an evaluation of their clarity and convincing nature:

1. **Performance Metrics**: The paper provides performance comparisons through tables that outline success rates for various robotic tasks. For instance, ARM4R shows a superior average success rate over other robotic models in specific tasks, such as picking and stacking cubes, achieving notable success rates of 96% for picking cubes and 61.3% for stacking cubes, which are higher than competitors like MVP, RPT, and others.
2. **Methodological Clarity**: The paper details the methodology behind the ARM4R model, including its unique approach of utilizing low-level 4D representations from human video data for pre-training. The transformation of 2D video representations into 3D point tracking through monocular depth estimation is explained, presenting a clear rationale for why this method is beneficial for robotic manipulation.
3. **Comparison with Existing Models**: The efficacy of ARM4R is further supported by comparative analyses, where it consistently outperforms baseline methods such as ATM and OpenVLA across various real-world tasks. These comparisons enable readers to gauge the relative improvements ARM4R offers, providing a compelling argument for its adoption.
4. **Generalization Across Tasks and Environments**: The paper claims that ARM4R’s shared geometric structure between 3D points and robot states facilitates better generalization across different robotic environments and tasks. This theoretical claim is underpinned by empirical results, which lend credence to the assertion.

**Essential References Not Discussed:**

I don’t find essential references that are not discussed.

**Experimental Designs Or Analyses:**

I have assessed the soundness and validity of the experimental designs outlined in the paper, especially focusing on the three training stages of the ARM4R model:

1. **Stage 1: Human Videos Pre-training**: The model is trained on a large dataset (76K videos from Epic-Kitchens100) that captures rich human-object interactions. This pre-training stage is crucial as it builds a foundation for understanding spatial dynamics and physical interactions. However, it would be beneficial to see a clear justification of the choice of dataset and its relevance to the targeted robotic tasks.
2. **Stage 2: Fine-tuning on 3D Point Tracking for Robotic Settings**: The model is fine-tuned on robotic videos, which is essential to adapt the learned representations to the robotic domain. The authors claim this stage only requires a modest amount of data (5-10% of Stage 1), but the criteria for selecting this data and whether it adequately captures the variations in robotic tasks should be justified. Additionally, potential discrepancies that may arise from differences in camera dynamics between human videos and robotic applications warrant further discussion.
3. **Stage 3: Fine-tuning for Robotic Control**: The final stage involves refining the model with a focus on actual robotic control tasks. The described methodology is reasonable, but revealing more about the range and variety of tasks included in this stage would strengthen the claims. Furthermore, a discussion on how fewer demonstrations were used compared to previous baselines could provide insights into the model's efficiency.

**Methods And Evaluation Criteria:**

The proposed method, ARM4R, presents a well-structured approach to enhance robotic model pre-training by leveraging low-level 4D representations extracted from human video data. The choice to use 3D point tracking converted from 2D video via monocular depth estimation effectively addresses the common challenge of limited large-scale datasets in robotics, which contrasts significantly with more data-rich domains like natural language processing and computer vision. This is a relevant application, as it aims to bridge the gap between human-centric visual data and robotic control tasks.

The evaluation criteria employed in the paper are also appropriate and rigorous. ARM4R was tested across multiple real-world robotic tasks, categorized into specific actions such as "pick," "stack," and "destack," which is essential for establishing a thorough understanding of the model's performance in practical scenarios. The success rates were reported for a variety of tasks, showing how ARM4R outperforms existing methods (like OpenVLA and ATM) that rely solely on robotic data, indicating a meaningful benchmark for comparison.

**Other Comments Or Suggestions:**

I don’t have other comments or suggestions.

**Other Strengths And Weaknesses:**

**Strengths:**

1. **Originality**: The paper introduces ARM4R, an Auto-regressive Robotic Model that utilizes low-level 4D representations derived from human video data. This approach represents a significant advancement in the application of representation learning to robotics, particularly by focusing on the transferability of learned representations from human actions to robotic control tasks. The novelty of leveraging dense 3D point tracking over time enhances the model's capability to generalize across various robotic platforms.
2. **Significance**: The work addresses a critical bottleneck in robotics—the lack of large-scale, diverse datasets suitable for training foundation models. By demonstrating that it is possible to pre-train on human video data, the paper opens new avenues for developing robotic models that can learn from the rich visual and action data available on the internet. This approach is particularly valuable for scenarios where collecting robotic data is either challenging or costly, thereby expanding the potential for scalable and data-efficient learning in robotics.
3. **Clarity**: The paper is well-structured, with a clear progression from the introduction of the problem to the methodology and results. The authors effectively communicate their contributions, and the use of relevant figures and diagrams enhances comprehension. Each section follows logically, making it easier for readers to grasp complex concepts associated with 4D representations and auto-regressive modeling.

**Weaknesses:**

1. **Limited Discussion on Limitations**: While the paper addresses some limitations, such as the coupling of object and camera motion affecting learned representations, it could benefit from a more comprehensive discussion regarding other possible weaknesses of the approach. For instance, greater consideration of potential biases in human video data, challenges in occlusion handling, and the model's performance on tasks with high variability could provide a more balanced perspective.
2. **Potential Domain Gaps:** Since there are domain gaps between human videos and robotic control data, discussion on how to address this gap effectively would help make the paper more convincing.

**Questions For Authors:**

1. How fewer demonstrations were used compared to previous baselines in Stage 3: Fine-tuning for Robotic Control? What is the relation between demonstrations number and fine-tuned model performance? How many demonstrations are needed for a good performance at least?
2. Does the domain gap between human videos and robotic control data have significantly influence on the model’s final performance?

**Relation To Broader Scientific Literature:**

The paper proposed to use human videos for pre-training which is not used widely in previous literature. And this paper shows pre-training with human videos could boost the model performance for robotic tasks.

**Theoretical Claims:**

The support of proposed method appears to be on empirical evaluations and quantitative comparisons rather than theoretical proofs. Thus, there are no specific theoretical claims or proofs outlined that were checked for correctness in the paper. There are extensive experiments showing the effectiveness of proposed method.

---

> ### Author Rebuttal · Authors · 2025-04-01
>
> We thank the reviewer for the insightful comments. We address the concerns raised below:
>
> 1. **Additional Limitations:** Here we provide a further analysis investigating more limitations of our approach.
>
>  ***(i) Unnatural rotation:*** We examined a new task, “put knife” in RLBench. Interestingly, this simulated task features an unnatural whole-arm rotation to grasp the knife handle, based on the expert demonstrations created by the simulator’s motion planning. As the movement between two key points is very long, we hypothesize that our model has difficulty learning this rotation.
>
>  ***(ii) Precision:*** We observed that ARM4R struggles with the 'screw bulb' task, a procedure requiring precise insertion into the bulb holder. In contrast, our model effectively performs other standard precision tasks, such as 'open drawer,' which necessitate only standard precision.
>
> 2. **High Variability (Dynamic Environments) task performance:** ARM4R can handle dynamic changes as it predicts each step in a trajectory, thus interacting with the environment continuously. To further evaluate dynamic variability, we test three cube tasks (stack, destack, and pick), manually shifting the cube during execution when the arm reaches ⅓ and ⅔ of its starting height. See the results:
>
> |Generalization to Dynamic Environments|pick|stack|destack|
> |----------------------------------------|--------------|--------------|--------------|
> |no movement| 96.0$\pm$2.3|61.3$\pm$1.3| 94.7$\pm$1.3|
> |cube moved when robot at ⅓ height|94.7$\pm$3.5|60.0$\pm$0| 93.3$\pm$1.3|
> |cube moved when robot at ⅔ height|92.0$\pm$2.3|57.3$\pm$1.3|90.7$\pm$1.3|
>
> Results show consistent performance despite target object movement, as long as sufficient z-axis space is available for adjustment. This highlights the model's robustness to environmental and object-level changes.
>
> 3. **Robustness analysis:** To assess the robustness of ARM4R regarding factors like camera noise, occlusions, or sensor drift, we perform additional evaluations on the three cube tasks under the following conditions: (1) Dim lighting: ambient light reduced to 50%, (2) Background distractors: dynamic background changes, such as people walking by or adjusting curtains, and (3) Tabletop distractors: two random objects placed near the target. The results are presented below.
>
> | Real-world Robustness|pick|stack|destack|
> |-----------------------|--------------|--------------|--------------|
> |standard|96.0$\pm$2.3|61.3$\pm$1.3|94.7$\pm$1.3|
> |dim light|86.7$\pm$1.3|52.0$\pm$2.3|81.3$\pm$2.7|
> |background distractor|94.7$\pm$1.3|57.3$\pm$1.3|90.7$\pm$1.3|
> |table-top distractors|81.3$\pm$1.3|48.0$\pm$2.3|74.7$\pm$1.3|
>
> Overall, ARM4R demonstrates robustness to lighting changes and background distractors, likely due to the attention pooling (See Fig 2 in the main paper), which guides the model to focus on the target region rather than background features. This is further supported by the observed performance drop when tabletop distractors are introduced.
>
> 4. **Dataset Bias:** We note that EpicKitchens is one of the main vision datasets containing various kinds of interactions between human hands and objects in kitchens. Thus, we choose to leverage it as a pre-training dataset for tabletop manipulation tasks. In future work, we plan to explore the benefits and effects of adding more pre-training datasets.
>
> 5. **Domain Gap:** As the reviewer correctly pointed out, there exists a domain gap between human video data and robotic control environments. To address this issue, we introduced in the paper Stage 2 of our training pipeline: Fine-tuning 3D Point Tracking for Robotic Settings. This stage uses the same task as in Stage 1 but enables the model to adapt to differences in camera dynamics and embodiment from the target robotic environment. The impact of this domain gap is evident in our results (Section 4.4, Figure 3), where omitting Stage 2 leads to decreased success rates. These findings underscore the importance of Stage 2 in bridging the gap and improving downstream performance.
>
> 6. **Number of demonstrations used:** In our real Kinova setting, we used 200 demonstrations (split into 19:1 training/validation sets) per task to finetune our model and all the other baselines except RPT and LLARVA in Tables 2 and 3. The RPT model uses 1920 demonstrations for its sensorimotor pre-training, and LLARVA uses 800 demonstrations per task. We note that for our newly added results on LLARVA, we used 200 demonstrations per task, following the same setting as the other baselines in Table 2. This may explain LLARVA’s poor performance on most of the tasks tested, as it typically requires more data. Overall, we found that training our model on 200 demonstrations is enough to achieve good performance in both real and simulation settings.

---

> > ### Comment · Reviewer_vJYs · 2025-04-05
> >
> > Thank you for the authors’ response to address my concern. I decided to keep my score on this paper.

---

### Official Review · Reviewer_Zx5c · 2025-03-12

**Overall Recommendation:** 3

**Summary:**

The paper introduces ARM4R (Auto-regressive Robotic Model with 4D Representations), a novel approach to pre-training robotic models using 4D representations derived from human video data. The key idea is to leverage 3D point (generated by depth estimation) tracking from human videos to create a low-level representation that captures the physical world's properties. This representation is then transferred to robotic control tasks, enabling the model to generalize across different robotic environments and configurations. ARM4R is pre-trained on large-scale human video datasets (e.g., Epic-Kitchens100) to learn spatial dynamics and physical interactions, and the training process can be split into three stages: stage1: pre-training on human video data to learn 3D point tracking, stage2: fine-tuning on robotic video data to adapt the model to the specific robotic setup, stage3: fine-tuning for robotic control tasks using proprioceptive data. ARM4R outperforms existing methods on both simulated (RLBench) and real-world robotic tasks, demonstrating superior generalization and transfer learning capabilities.

**Claims And Evidence:**

The claims are supported by clear and convincing evidence.

**Essential References Not Discussed:**

None

**Experimental Designs Or Analyses:**

The RLBench setting and real-world experiments make sense.

**Methods And Evaluation Criteria:**

Methods and evaluation criteria make sense.

**Other Comments Or Suggestions:**

None

**Other Strengths And Weaknesses:**

### Strengths
1. ARM4R demonstrates that pre-training on human video data can lead to better performance than models pre-trained solely on robotic data.
2. ARM4R shows strong generalization capabilities across different robotic platforms (e.g., Kinova to Franka).

### Weakness
1. Lack of comparison with more competitive methods in RLBench, like RVT [1], RVT-2 [2].

[1] Goyal, Ankit, et al. "Rvt: Robotic view transformer for 3d object manipulation." Conference on Robot Learning. PMLR, 2023.
[2] Goyal, Ankit, et al. "Rvt-2: Learning precise manipulation from few demonstrations." arXiv preprint arXiv:2406.08545 (2024).

**Questions For Authors:**

1. How to acquire the point cloud for tracking training, which depth estimation model do you use? Are there any influences of using depth estimation that lacks absolute depth information?

**Relation To Broader Scientific Literature:**

None

**Theoretical Claims:**

The equations seem correct.

---

> ### Author Rebuttal · Authors · 2025-04-01
>
> We thank the reviewer for the insightful comments. We address the concerns raised below:
>
> 1. **Comparison to RVT, RVT-2:** As the reviewer noted, RVT and RVT-2 are recent transformer-based methods that predict keyframes in robot trajectories. Importantly, both methods leverage RGB-D images to reconstruct a scene's point cloud and then predict keypoints. This point cloud reconstruction relies on a similar principle to our method, with ARM4R using pre-training on the 3D point tracking task to learn scene structure. We will add a proper discussion and comparison to this line of work in the final version of the paper.
>
> However, we would like to highlight that our approach contains some significant differences compared to RVT/RVT-2. First, our approach leverages 3D point tracking from human video using depth estimation, relying on pseudo labels. This enables the utilization of readily available web-based video data. In contrast, RVT and RVT2 depend on ground-truth RGB-D data and reconstructed point clouds captured by cameras. This dependence on ground-truth depth data inherently restricts the amount of usable video data. In addition, our approach uses 3D point tracking that can be easily extracted from human video data compared to keyframe-based approaches which leverage heuristics such as velocity changes to find keyframes.
>
>
> 2. **Depth Estimation Model:** To obtain the 3D point tracks, we use SpatialTracker, which utilizes ZoeDepth [1] as the depth estimator. We note that since ZoeDepth predicts metric depth, it also predicts the absolute depth map. Additionally, as demonstrated in TapVid-3D [3]—a follow-up work of SpatialTracker—and DELTA [4], utilizing UniDepth [2], a more accurate monocular depth model, can enhance tracking performance. Lastly, as mono-depth methods further improve (such as UniDepth v2 [5]), 3D point tracking on monocular videos will also further improve.
>
> [1] Bhat, Shariq Farooq, et al. "Zoedepth: Zero-shot transfer by combining relative and metric depth." arXiv preprint arXiv:2302.12288 (2023).
>
> [2] Piccinelli, Luigi, et al. "UniDepth: Universal monocular metric depth estimation." Proceedings of the IEEE/CVF Conference on Computer Vision and Pattern Recognition. 2024.
>
> [3] Koppula, Skanda, et al. "Tapvid-3d: A benchmark for tracking any point in 3d." arXiv preprint arXiv:2407.05921 (2024).
>
> [4] Ngo, Tuan Duc, et al. "DELTA: Dense Efficient Long-range 3D Tracking for any video." arXiv preprint arXiv:2410.24211 (2024).
>
> [5] Piccinelli, Luigi, et al. "Unidepthv2: Universal monocular metric depth estimation made simpler." arXiv preprint arXiv:2502.20110 (2025).

---

### Official Review · Reviewer_aq5E · 2025-03-15

**Overall Recommendation:** 3

**Summary:**

The paper proposes to pre-train the vision-language-action models using the sparse 4D trajectories (obtained as pseudo-labels using pre-trained networks) of humans performing actions in ego-centric videos. Such pre-training is claimed to be helpful when the model is fine-tuned on the downstream task of action trajectory prediction. The paper is well written. The idea presented in the paper is interesting and well-motivated, and the aimed contributions are timely. The main problem of the paper is that the experimental results are not convincing.  Thus, the offered conclusion could be misleading.

**Update** I have upgraded my rating to weak accept. Thank you for the rebuttal.

**Claims And Evidence:**

No. The experimental results arguably support the claimed contributions, For example:

1. In Table 1, the major gain is due to one single action, "put money". It is unclear if such high success, for one particular case, is merely due to statistical insignificance or due to the nature of that particular action. Otherwise, the proposed method does not seem to be helpful, contrary to the claim of the paper.

2. Table 2 has limited methods, which obscures the complete picture.

3. Table 4 does not show what happens if Epic is not used but Kinova is used.

**Essential References Not Discussed:**

None.

**Experimental Designs Or Analyses:**

The experimental designs are valid. The paper follows the standard evaluation protocol.

**Methods And Evaluation Criteria:**

The proposed method and evaluation criteria are meaningful. The issue is not on this part.
Instead, the problem stems from the fact that the statistical significance of the experimental results is questionable.

**Other Comments Or Suggestions:**

The details of the human dataset preparation are missing. The fact that the human videos consist of distractors, moving camera/head, and bimanual manipulations, they may need to be taken into the account.

**Other Strengths And Weaknesses:**

The presentation of the paper and the visualization within are of high quality and helpful.

**Questions For Authors:**

Please provide more details of the data preparation and validate your claims through more rigorous experimental results.

**Relation To Broader Scientific Literature:**

The paper aims to make a timely contributions.

**Theoretical Claims:**

There is no theoretical claim that needs to be checked.

---

> ### Author Rebuttal · Authors · 2025-04-01
>
> We thank the reviewer for the insightful comments. We address the concerns raised below:
> 1. **Table 1 Results:** As noted by the reviewer, the simulation results for ARM4R are slightly skewed by the high success rate for the “put money” task. We recalculate the success rate without this outlier for ARM4R and the closest competitor, PerAct: Our model achieves a success rate of 56.51%, while Peract achieves a marginally lower success rate of 56.36%. We note that PerAct also has access to ground truth voxel information from the simulator, while our model learns 3D understanding from its own training, only using RGB images at inference time.
>
> To further evaluate our model against other 3D-based approaches, we also compare it with 3D-VLA—a 3D-based VLA model that generates goal point clouds. Since the official training and inference codebase is not publicly available, we compare ARM4R to 3D-VLA using three RLBench tasks for which results are reported in the 3D-VLA paper. The results are shown as follows:
>
> |Method|put knife|pick up cup|take umbrella|Average|
> |--------------------|-----------|-------------|----|-------------|
> |Image-BC ViT|0|4|0|1.33 |
> |C2FARM-BC|4|28|48|26.67|
> |LLARVA|4|52|32| 29.33|
> |PerAct|20|56|76|50.67|
> |3D-VLA|68|40|52|53.33|
> |**ARM4R**|28.8±1.50|99.2±0.80|82.4±2.03|70.13±1.53|
>
> We see that ARM4R performs much better than all other tested models on 2/3 tasks, and the average performance is better as well. The poor performance on the put knife task is explained by the nature of data collected. This simulated task features an unnatural whole-arm rotation to grasp the knife handle, based on the expert demonstrations created by the simulator’s motion planning. Due to the large movement between the first two keyframes, we hypothesize that our model struggles to learn the rotation because the context window becomes saturated with local motion details, leading to a loss of global trajectory information. This is why LLARVA struggles with this task as well, since it has to predict the whole trajectory without skipping any step’s action. In contrast, 3D-VLA predicts keyframes and thus does not suffer from this issue. Finally, since our model is pre-trained on human data and this rotation is very unnatural, it might be more difficult to learn.
>
> 2. **Table 2 Results:** We add two more baselines to our real Kinova setting: (1) LLARVA, a 2D-based VLA, and (2) Pi0-FAST, a recent SoTA VLA model. The results are below:
>
> |Method|pick yellow|pick cyan|pick green|destack yellow|destack cyan|stack yellow|stack cyan|
> |:-----------:|:--------------:|:--------------:|:---------------:|:---------------:|:---------------:|:---------------------------------:|:--------------:|
> |LLARVA|44.4 $\pm$ 6.4|41.6 $\pm$ 4.2|54.2 $\pm$ 11.0|8.3 $\pm$ 4.8|10.3 $\pm$ 2.6|5.6 $\pm$ 2.8|12.8 $\pm$ 2.6|
> |Pi0-FAST|63.0 $\pm$ 3.7|33.3 $\pm$ 8.3|25.0 $\pm$ 0.0|5.5 $\pm$2.8|25.6 $\pm$ 2.6|22.2 $\pm$ 2.8|25.6 $\pm$6.8|
> |**Method**|**pick spiderman**|**pick penguin**|**pick pig**|**play basketball**|**push red button**|**push red button, then blue button**|
> |LLARVA|9.3 $\pm$ 1.3|9.3 $\pm$ 1.3|8.0 $\pm$ 2.3|10.7 $\pm$ 3.5|20.5 $\pm$ 5.1|2.8 $\pm$ 2.8|
> |Pi0-FAST|16.0 $\pm$ 2.3|17.3 $\pm$ 1.3|9.3 $\pm$ 2.7|13.3 $\pm$ 3.5|20.5 $\pm$ 2.6|0.0 $\pm$ 0.0|
>
> Both models exhibit low performance: LLARVA / Pi0 achieves an average success rate of 18.3% / 21.2%. This is likely due to the amount of fine-tuning data used: LLARVA typically requires 800 episodes per task (we have here 13 tasks), Pi0-FAST shows good results when tuned for the whole DROID dataset (~76K demos), while we only have a total of 2,600 training episodes for our real setting. For robotics, collecting a large amount of demonstrations is expensive and difficult, so learning efficiency is a key performance indicator.
>
> 3. **Table 4:** We added an extra experiment for Franka robot which includes: (1) Kinova 3D points tracking stage (2) Franka robot joint fine-tuning. The results are shown as follows:
>
> |pretrain|finetune|robot|pick|stack|destack|
> |----------|----------|--------|--------------|--------------|--------------|
> |-|Kinova|Franka|81.3$\pm$1.3|52.0$\pm$2.3|73.3$\pm$2.7|
>
> The results show that only training on Kinova 3D points tracking can also boost results when a Franka robot is used in the downstream setting, although the increase in performance is not as big as the one compared to when human video pre-training is also used. This aligns with the hypothesis that both Stage 1 and Stage 2 are important to achieve the best possible performance.
>
> 4. **Dataset preparation:** We provide details on human video dataset preparation in Appendix B1. Regarding camera motion, static cameras are a subset of dynamic cameras; thus, our robotic settings are not out of distribution with respect to the pre-training.
>
> 5. **Additional Experiments:** Due to space constraints, please refer to points 2 and 3 in response to reviewer vJYs for experiments on handling distractors and dynamic environments.

---

### Decision · Program_Chairs · 2025-05-01

**Decision:**

Accept (poster)

**Comment:**

The paper proposes a method for robotic manipulation learning from human video via 3D point tracking in time, which relies on monocular depth estimation from 2D images. The evaluation is done in RLBench and on physical robot setups.

The reviewers found merits in the method's motivation and technical approach. At the same time, initially they also had a variety of concerns, ranging from the evaluation on RLBench being inadequate to lack of clarity around handling the embodiment gap between people and robots, to potential bias in the pretraining tasks, which are geared towards kitchen environments.

Overall, the authors have successfully addressed these concerns in the rebuttals, partly by adding significantly more experiments and analysis during the rebuttal stage and partly by carefully positioning the proposed method, ARM4R, w.r.t. the related work. The metareviewer encourages the authors to incorporate the results and explanations from their rebuttals into the paper and has no major concerns about accepting this work.